# HIV-1 capsids enter the FG phase of nuclear pores like a transport receptor

Liran Fu[1,4], Erika N. Weiskopf[2,4], Onno Akkermans[2,4], Nicholas A. Swanson[2], Shiya Cheng[3], Thomas U. Schwartz[2✉] & Dirk Görlich[1✉]

HIV-1 infection requires nuclear entry of the viral genome. Previous evidence suggests that this entry proceeds through nuclear pore complexes (NPCs), with the 120 × 60 nm capsid squeezing through an approximately 60-nm-wide central channel[1] and crossing the permeability barrier of the NPC. This barrier can be described as an FG phase[2] that is assembled from cohesively interacting phenylalanine–glycine (FG) repeats[3] and is selectively permeable to cargo captured by nuclear transport receptors (NTRs). Here we show that HIV-1 capsid assemblies can target NPCs efficiently in an NTR-independent manner and bind directly to several types of FG repeats, including barrier-forming cohesive repeats. Like NTRs, the capsid readily partitions into an in vitro assembled cohesive FG phase that can serve as an NPC mimic and excludes much smaller inert probes such as mCherry. Indeed, entry of the capsid protein into such an FG phase is greatly enhanced by capsid assembly, which also allows the encapsulated clients to enter. Thus, our data indicate that the HIV-1 capsid behaves like an NTR, with its interior serving as a cargo container. Because capsid-coating with *trans*-acting NTRs would increase the diameter by 10 nm or more, we suggest that such a 'self-translocating' capsid undermines the size restrictions imposed by the NPC scaffold, thereby bypassing an otherwise effective barrier to viral infection.

To establish infection, retroviruses must integrate a DNA copy of their reverse transcribed RNA genomes into a host chromosome. The nuclear envelope (NE) is thus a barrier to retroviral infection. Most retroviruses rely on mitotic NE breakdown for nuclear entry and therefore only infect proliferating cells. However, lentiviruses, such as HIV-1, are exceptional in that they infect non-dividing cells with intact NEs. Here, passage through nuclear pore complexes (NPCs) is an obligatory step.

NPCs have a mass of about 100 MDa and provide a channel for nucleocytoplasmic transport[4,5] which is controlled by a permeability barrier[2,6]. Nuclear transport receptors (NTRs) enable active transport, cross NPCs in a facilitated manner and circulate between the nucleus and cytoplasm[7–10]. Members of the importin β superfamily represent the largest NTR class. They draw energy from the RanGTPase system, bind cargoes in a RanGTP-controlled fashion and translocate them across the NPC barrier. NPCs are built from about 30 different nucleoporins (Nups), including about ten so-called FG-Nups that anchor barrier-forming phenylalanine–glycine (FG) repeat domains to the NPC scaffold.

FG domains have low sequence complexity, are intrinsically disordered and harbour numerous FG dipeptide motifs which bind NTRs during facilitated translocation[11–14]. To explain how this interaction favours the NPC passage of NTRs, several models have been proposed, including the 'reduction in dimensionality model'[15], the 'affinity gradient model'[16], the 'Brownian affinity gate model'[17] and models that see FG domains as (non-interacting) entropic brushes[18].

By contrast, the 'selective phase model'[6] considers that FG repeats can also confer multivalent cohesive interactions and reversibly cross-link FG domains to a sieve-like FG phase[19–22]. It assumes that NTRs and NTR–cargo complexes 'melt' through such a phase by binding and competing inter-FG-repeat interactions, whereas inert macromolecules are rejected by the sieve structure, unless NTRs recognize them as valid cargo.

The FG domain of Nup98 and its homologues[12,23] are special in that they occur in very high copy numbers and contribute the most FG mass per NPC[24]. They have the highest count (about 50) and density (about one per 12 residues) of FG motifs per domain. Water is a poor solvent for these FG domains; therefore, they readily phase-separate from dilute aqueous solutions to form a very protein-dense (about 400 mg ml$^{-1}$) FG phase[21,25]. The resulting cohesive FG phases recapitulate nuclear transport selectivity very well, fully excluding inert molecules such as GFP or mCherry while allowing entry of NTRs and their cargo complexes with very high partition coefficients. Furthermore, the partition coefficient of a mobile species in a cohesive FG phase is an excellent predictor of its NPC passage rate, with the two parameters correlating well over a range of four orders of magnitude[26]. Such an in vitro reconstituted FG phase thus provides a convenient way to study the properties of the NPC barrier and its interactions with NTRs.

Early steps in HIV-1 infection, namely surface receptor binding and membrane fusion, deliver the viral capsid into the cytoplasm of

[1]Department of Cellular Logistics, Max Planck Institute for Multidisciplinary Sciences, Göttingen, Germany. [2]Department of Biology, Massachusetts Institute of Technology, Cambridge, MA, USA. [3]Department of Meiosis, Max Planck Institute for Multidisciplinary Sciences, Göttingen, Germany. [4]These authors contributed equally: Liran Fu, Erika N. Weiskopf, Onno Akkermans. ✉e-mail: tus@mit.edu; goerlich@mpinat.mpg.de

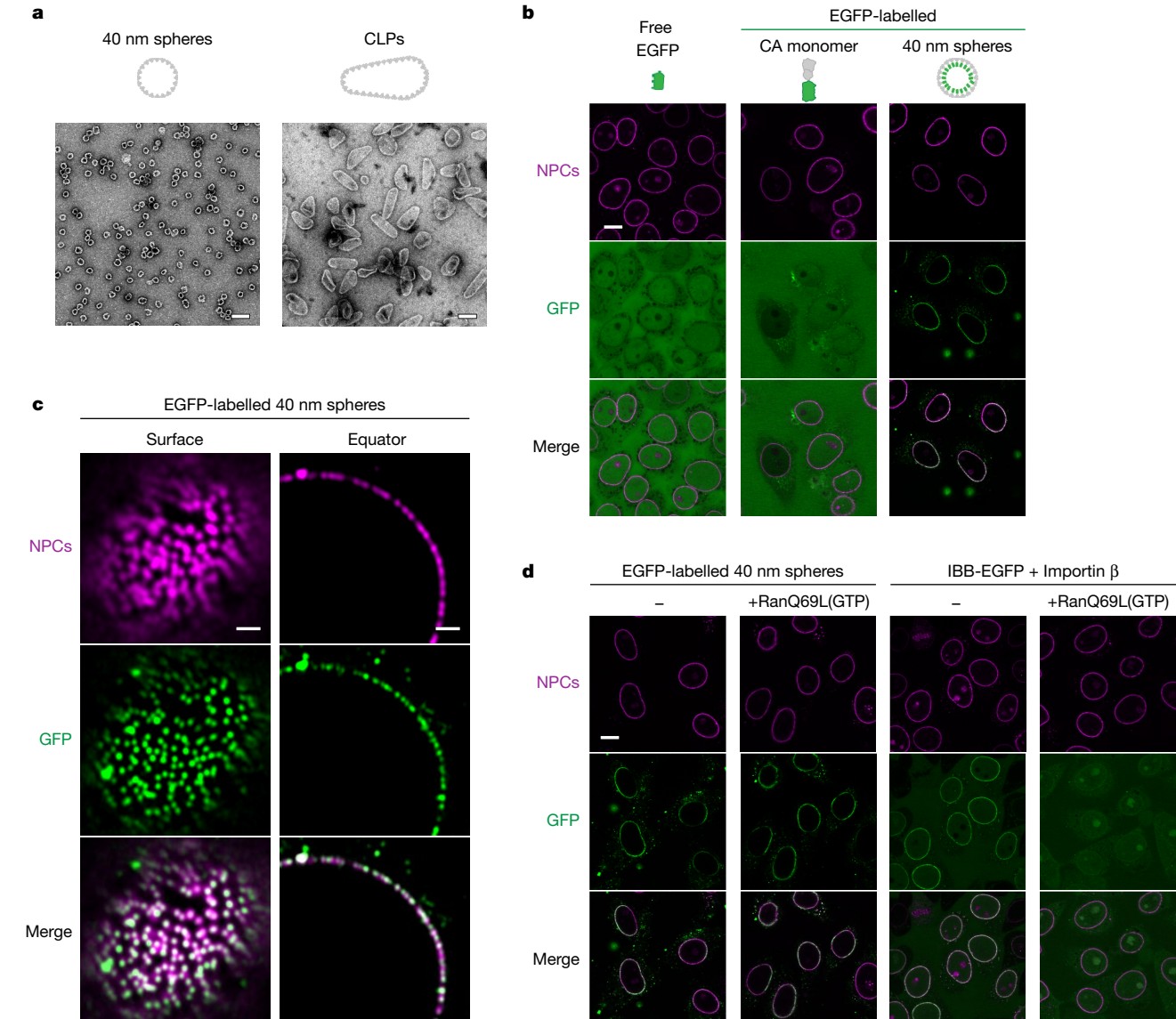

**Fig. 1 | Autonomous targeting of HIV-1 capsid species to NPCs.**
**a**, Representative negative-stain electron micrographs of HIV-1 capsid preparations used in this study. **b**, HeLa cells were grown in multiwell slides and permeabilized with 30 μg ml⁻¹ of digitonin to perforate their plasma membranes[6,46], releasing soluble transport factors and allowing entry of the indicated fluorescent species: EGFP (3 μM), a monomeric CA–EGFP fusion (200 nM) or 40 nm capsid spheres (about 2.8 nM with 15% of the CA protomers being fused to a C-terminal GFP). After 30 min of incubation at room temperature, confocal laser scans were taken directly through the live samples. Note that the assembled capsid spheres bound very efficiently to NPCs and colocalized with the NPC marker (an Alexa647-labelled anti-Nup133 nanobody[47]). The monomeric CA–EGFP fusion gave no discernible NPC signal. Scan settings were adjusted individually. **c**, Targeting of 40 nm capsid spheres to NPCs was performed as in **b** but higher resolution images were taken by Airy scans on a Zeiss LSM880 microscope. **d**, Experiments were performed as in **b** but included control incubations with 3 μM RanGTP which was locked in its GTP state by a Q69LΔC double mutation. RanGTP displaces cargo from importins, which normally happens inside nuclei. The mutant, however, triggers this dissociation prematurely in the cytoplasm. The RanGTP resistance of capsid binding to NPCs indicates importin-independent targeting. By contrast, the NPC targeting of the IBB–GFP·importin β complex (0.2 μM) was completely abolished by the Ran mutant. Scan settings were identical for corresponding ±RanGTP pairs. Experiments were repeated independently with identical outcomes (**b**, *n* = 7; **c**,**d**: *n* = 3). Scale bars, 100 nm (**a**), 10 μm (**b**,**d**), 1 μm (**c**).

the target cell[27]. The capsid is composed of about 1,500 capsid protein (CA) molecules, arranged in approximately 250 hexameric and 12 pentameric capsomeres[28]. The capsid encloses two copies of genomic RNA (chaperoned by the NC protein), as well as the initially required viral enzymes, reverse transcriptase and integrase[29]. It has long been thought that the capsid uncoats in the cytoplasm. One argument was that previous structural models of the NPC had a central channel width of only 40 nm (refs. 30,31)—too narrow to accommodate an intact cone-shaped HIV-1 capsid of 60 × 120 nm. This concept has recently been challenged by studies indicating capsid uncoating inside the

nucleus[32,33], by new in situ NPC structures showing a channel that is around 60 nm wide[34,35] and by electron tomographic reconstructions of an HIV-1 capsid trapped in the central NPC channel[1]. However, how the capsid partitions into the FG phase remained unclear. If it were carried by NTRs, as are conventional cargoes, the extra NTR layer would increase the effective diameter of the capsid, making passage through the constrained NPC scaffold even less likely. This problem is well illustrated by the fact that the HIV-1 capsid alone is much larger than the experimentally determined approximately 36 nm size limit for NTR-mediated cargo transport[30].

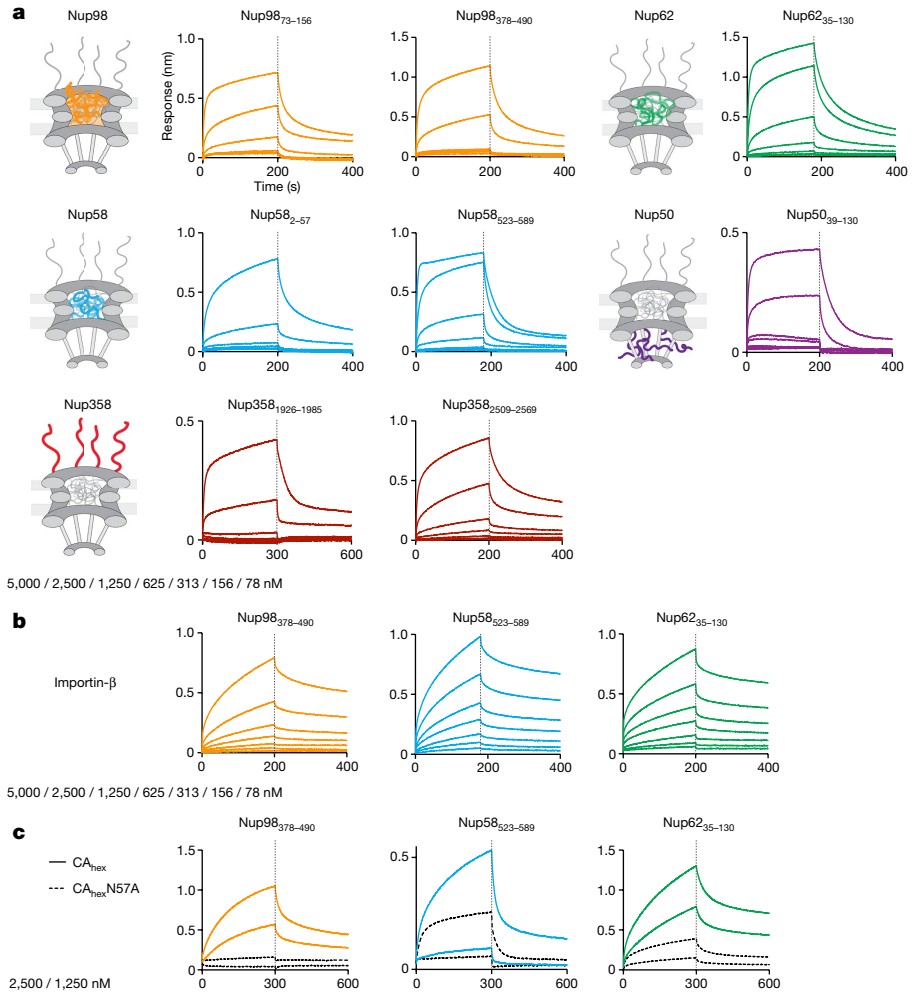

5,000 / 2,500 / 1,250 / 625 / 313 / 156 / 78 nM

5,000 / 2,500 / 1,250 / 625 / 313 / 156 / 78 nM

2,500 / 1,250 nM

**Fig. 2 | CA hexamers are general FG repeat binders. a**, BLI experiments to test binding of CA hexamers to various FG-domain fragments that had been immobilized on BLI sensor tips. Sensorgrams show time courses of binding and subsequent dissociation of disulfide-stabilized CA hexamers as an analyte. CA hexamer concentration steps are indicated. The FG-domain fragments were chosen to sample domains along the NPC transport channel. Rapid and dynamic interaction was observed with all FG probes, although quantitative differences are evident. **b**, Control experiments with importin β as the analyte

show that it binds indiscriminately to three different FG-domain fragments, as expected for an NTR. CA hexamer mimics this binding behaviour. **c**, Control experiment with CA hexamers and CA-N57A mutant hexamers binding to three different FG-domain fragments. The CA-N57A mutation is known to strongly reduce the binding to the established Nup153 and CPSF6 FG peptides at the FG-binding pocket on the CA hexamer[36,40]. The mutation also abolished CA hexamer binding to the Nup98 FG repeats but had only a moderate effect on the interaction with Nup62 and Nup58 FG repeats (dashed curves).

## Importin-independent NPC targeting of HIV-1 capsids

We reasoned that a solution to this conundrum might be related to the observation that the CA protein binds specialized FG motifs in the nuclear RNA polyadenylation factor CPSF6 (ref. 36) and Nup153 (ref. 37), enhanced through hydrophobic binding pockets created by hexamerization[38–40] and further augmented when binding to mature HIV-1 lattices[41]. While these specific FG units in CPSF6 and Nup153 have been explored in great detail, other studies have also indicated interactions with further FG-Nups[37,42–44]. Therefore, it seemed possible that FG binding of the capsid might allow it to 'melt' into the FG phase through an NTR-like mechanism, eliminating the need for assistance from host NTRs.

To test this idea, we assembled 40 nm capsid spheres from CA capsomeres[45] (labelled at 1:6 molar ratio with a GFP tracer fused to the CA carboxy terminus and pointing to the capsid interior). Electron microscopy confirmed their correct assembly (Fig. 1a). We incubated these fluorescent assemblies with digitonin-permeabilized HeLa cells[6,46] and observed efficient and clean targeting to NPCs, as indicated by colocalization of the capsid–GFP signal with an anti-Nup133 NPC marker[47]

(Fig. 1b,c). This NPC targeting required the assembly of the CA protein into a capsid (Fig. 1b) but not the addition of NTRs, even though NTRs are depleted from permeabilized cells[48]. Furthermore, the capsid targeting to NPCs was resistant to the RanQ69L (GTP) mutant (Fig. 1d, left), which is locked in its GTP state and prematurely displaces cargo from importins before import can occur[48]. This excludes importins as capsid NTRs and suggests that the capsid has self-translocating properties. The corresponding control, in which the importin β-dependent NPC targeting of an IBB–GFP fusion is blocked by RanQ69L(GTP), is shown in Fig. 1d (right).

## General, direct FG repeat interactions of capsomeres

To probe for direct interactions between CA and FG domains, we used biolayer interferometry (BLI) with disulfide-stabilized CA hexamers[49] as the analyte and various biotinylated FG regions immobilized on streptavidin sensor chips. We tested eight FG regions originating from five different Nups (Nup98, 62, 58, 50 and 358), encompassing 60–110 residues with several FG-dipeptides each and representing diverse types of FG repeats: with GLFG and FxFG motifs and both cohesive

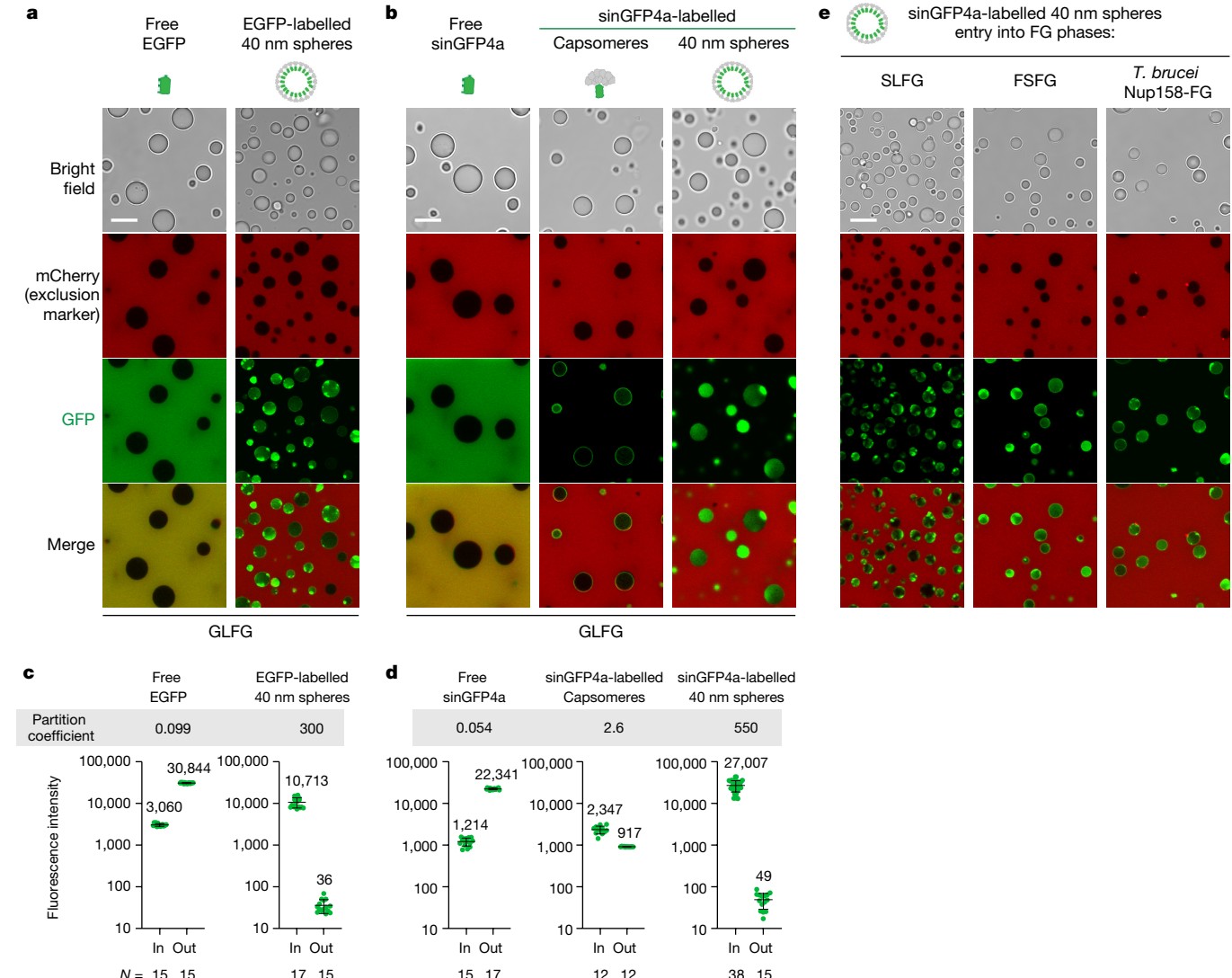

**Fig. 3 | Efficient FG phase entry of HIV-1 capsid spheres. a,b**, FG phases were assembled by phase-separating a well-characterized, cohesive GLFG repeat domain (ref. 50; Supplementary Fig. 1 for FG-domain sequences) and then probed for entry of indicated fluorescent species. Non-fused GFPs and mCherry remained firmly phase-excluded, whereas 40 nm capsid spheres accumulated to high partition coefficients. Partially assembled (hexameric) capsomere–sinGFP4a fusions showed only faint surface binding (visible in high-sensitivity scans). Phase separation was initiated by rapidly diluting FG domains (1 mM in 2 M guanidinium·HCl) with 25 volumes of buffer (50 mM Tris/HCl pH 7.5, 250 mM NaCl, 1 mM IP6), followed 5 min later by a further fourfold dilution in buffer with indicated fluorescent probes. Confocal scans were taken after another hour, when the FG particles had settled to the slide bottom. Fluorescent probes: mCherry (3 µM), free EGFP or sinGFP4a (3 µM), capsomeres (0.4 µM), 40 nm capsid spheres (10 nM, with 15% EGFP- or sinGFP4a-labelled CA protomers).

Scan settings were adjusted individually. **c,d**, Quantification of **a,b** but with larger datasets. For each datapoint (representing one FG particle), the integrated intraparticle signal (in) was compared to outside regions (out). Note the log-linear scale and that free GFP derivatives are well excluded. Capsid spheres accumulated to very high partition coefficients (in:out = 300–500) which are still underestimated because the outside signals were as low as the background (background fluorescence + instrumental noise) and this background was not subtracted (to avoid division by zero). The 'in-value' for the capsomere partitioning refers to the intraphase signal, excluding the rim. Numbers are means; bars indicate mean ± s.d. $N$ = number of quantified FG particles and outside areas, respectively. **e**, FG phase entry of free mCherry and 40 nm capsid spheres was probed as in **a,b** but using SLFG repeats, FSFG repeats or FG repeats from *T. brucei* Nup158. Experiments were repeated with identical outcomes (**a**, $n$ = 4; **b**, $n$ = 13; **e**, $n$ = 4). Scale bars, 10 µm.

and non-cohesive repeats (Extended Data Fig. 1). The recorded sensorgrams (Fig. 2) showed rapid and dynamic binding of CA hexamers to all tested FG repeats—qualitatively similar to the specialized FG motifs of the Nup153 and CPSF6 controls (Extended Data Fig. 2), although with noticeable individual variations. The binding was specific, as judged by reductions in interaction with the CA-N57A mutation (Fig. 2c), which is known to impede the interaction with FG motifs from Nup153 and CPSF6 (refs. 36,40) (Extended Data Fig. 2a). The observed FG binding of hexameric CA was also qualitatively similar to that of the prototypic NTR, importin β (Fig. 2b). Thus, the FG interaction sites of a CA hexamer

confer NTR-like and apparently multivalent interactions with a wide range of FG repeats.

## Autonomous FG phase entry of HIV-1 capsid species

To assess whether these FG interactions are productive in terms of facilitated translocation, we next asked whether the capsid assemblies were able to enter the permeability barrier and specifically into an in vitro reconstituted FG phase (Fig. 3). For the initial experiments, we used a well-characterized FG domain comprising 52 GLFG 12mer

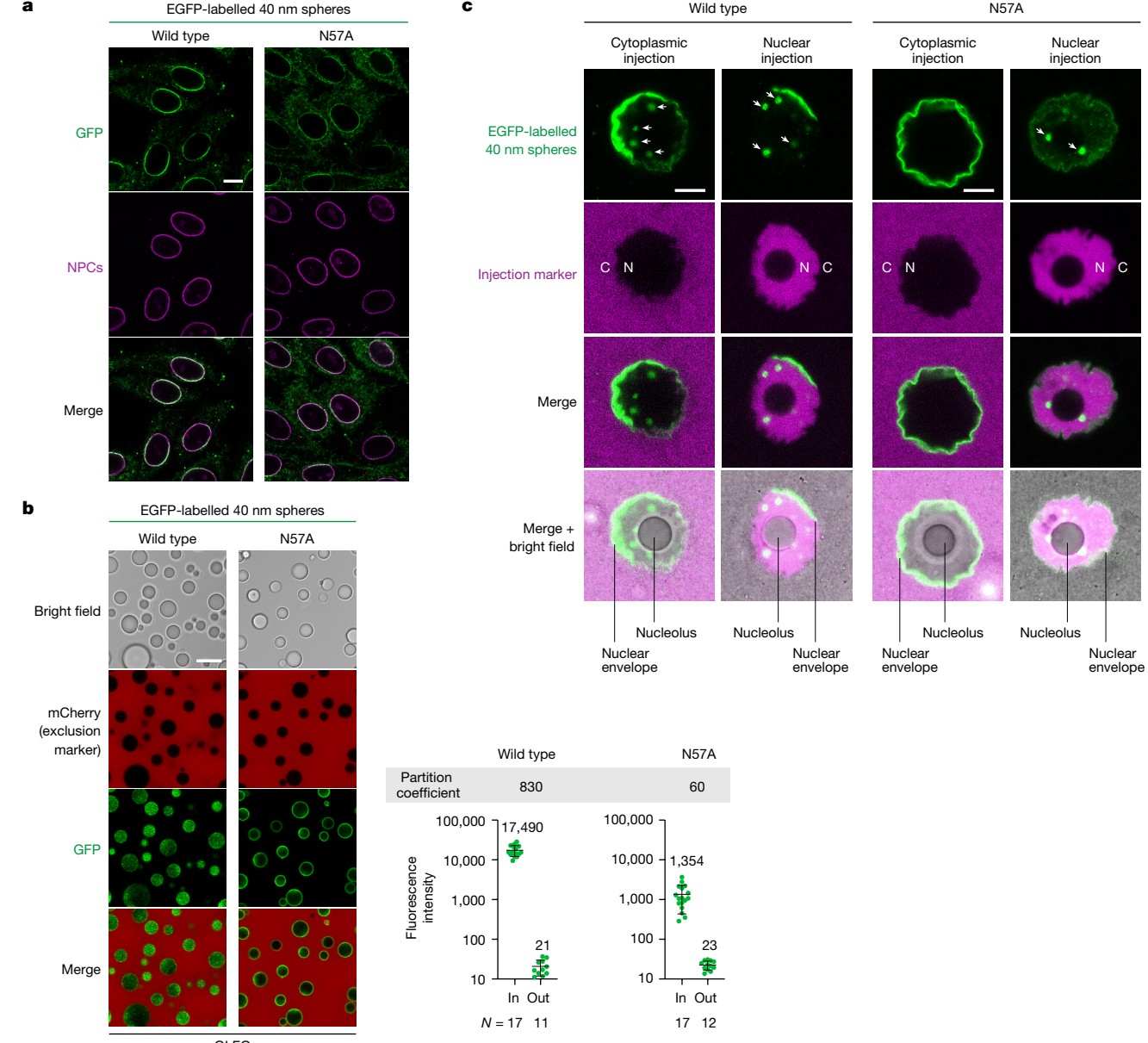

**Fig. 4 | The CA-N57A mutation impedes capsid passage through NPCs.**
**a**, Targeting of capsid spheres (4 nM; wild type or CA-N57A mutant) to HeLa
cell NPCs was tested as in Fig. 1. The mutation reduced the capsid signal at
NPCs. **b**, The 40 nm capsid spheres were tested for partitioning into the GLFG
phase as described in Fig. 3. The wild-type spheres entered the FG phase
completely. CA-N57A mutant spheres, however, bound only to the surface of
the phase, indicating that their FG interactions are not strong enough to melt
cohesive, barrier-forming interactions in the FG phase. Numbers are means;
bars indicate mean ± s.d. $N$ = number of quantified FG particles and outside
areas, respectively. **c**, The 40 nm capsid spheres (wild type or N57A) were
injected (together with a tetrameric tCherry) into either the cytoplasm or the
nucleoplasm of fully grown mouse oocytes. Images show confocal laser scans
25 min after injection, after reaching the endpoint distributions. Cytoplasmically
injected wild-type capsids not only bound to the NE but also traversed NPCs

and accumulated in intranuclear foci (indicated by white arrows); the tCherry
injection marker remained cytoplasmic, indicating proper injections and
intact NE and NPC barriers. N57A mutant spheres enriched only at the NE but
failed to complete NPC passage and to reach the nuclear interior. The nuclear
injection confirmed that the mutant would have bound to the intranuclear
foci if it had reached the nucleoplasm. Images show the full nucleus and its
surrounding cytoplasm. Nuclear (N) and cytoplasmic (C) regions can be
distinguished by the injection marker (in magenta). NE and nucleolus are
indicated by lines in the merged images. See Extended Data Fig. 7 for $z$-stacks
and Extended Data Fig. 6 for a complete oocyte imaged after cytoplasmic
injection of wild-type 40 nm spheres. Scan settings were identical for each wild
type and N57A mutant pair. Experiments were repeated independently with
identical outcomes (**a**,**b**,**c**: $n$ = 3). Scale bars, 10 μm.

repeat units[50] (with the sequence GGLFGGNTQPAT; Supplementary
Fig. 1). It represents a generic FG phase model, originally derived by
the regularization of a ciliate Nup98 FG domain, which excludes any
specific adaptations of HIV-1. We also chose this model because it
avoids complications such as $O$-glycosylation or amyloid formation
and because it is very well characterized and known to recapitulate

importin- and exportin-mediated cargo transport, response to the
RanGTPase system, as well as NTF2-mediated retrieval of RanGDP to the
nucleus[50].

In all experiments, we first initiated the (instantaneously occurring)
phase-separation reaction and then added fluorescent permeation
probes. As expected, mCherry (26 kDa, diameter 5 nm) remained firmly

excluded from the phase with a very low partition coefficient (in:out) of 0.05 or less (Fig. 3a,b). The 40 nm capsid spheres, however, completely entered the FG phase, accumulating to a partition coefficient of 300 or more with essentially no signal above background remaining outside the phase (Fig. 3a,c). Consistent with smaller particles having a larger relative surface area for absorption, intraphase accumulation was higher in smaller FG particles, indicating also that intraphase diffusion was rate-limiting. Nevertheless, the 40 nm capsid spheres reached the centre of about 5 μm-sized FG particles within 1 h of incubation (Fig. 3a), which corresponds to a diffusion distance more than ten times longer than the transport path through NPCs. Given that transport time scales with the square of the distance, it can be estimated that the actual capsid entry into the NPC could happen within 1 min if only the partitioning into the FG phase were rate-limiting.

NPCs have long been known to act as sieves, with exclusion scaling with the size of the mobile species[51]. Given that the spheres are very large in mass (6 MDa) and diameter (40 nm), their efficient FG phase entry may seem surprising. However, this can be explained by (1) the cooperation of hundreds of FG-binding sites on the surface of the homopolymeric capsid spheres and (2) by the burial of FG-repellent elements[26], such as the relatively charged CA 'interior' and the C-terminally fused GFP.

In support of this, we observed that FG phase entry of the capsid remained highly efficient when the standard GFP was replaced by the super-inert (and thus highly FG-repellent) sinGFP4a variant[26] (Fig. 3b and Extended Data Fig. 3). This seems to be possible only when sinGFP4a is completely enclosed by the capsid and thus not in contact with the FG phase. Indeed, the control with sinGFP4a-tagged hexameric capsomeres (with one exposed sinGFP4a) showed only a distinct binding to the phase surface and failed to enter the phase efficiently (Fig. 3b and Extended Data Fig. 3), even though this species is 30 times smaller than the capsid spheres.

The capsid spheres also entered other types of FG phases, such as a cohesive phase with SLFG or FSFG motifs[25] (Fig. 3e), which represent different, common FG types. Likewise, we observed efficient capsid partitioning into the FG phase of human Nup98 itself and the *Saccharomyces cerevisiae* Nup98 homologue Nup116 (Extended Data Fig. 4). Note that the hNup98 FG domain lacked the modulating *O*-GlcNAc modification[52], resulting in a phase that was hypercohesive, allowing only very effective translocators to enter. The capsid spheres thus belong to this category of translocators.

Moreover, the capsid spheres efficiently partitioned into an FG phase derived from an evolutionarily very distant Nup98 homologue, *Trypanosoma brucei* Nup158 (Fig. 3e), whose FG domain is dominated by less hydrophobic GFG motifs that occur at a higher-than-usual density[21]. In all cases, complete phase entry seemed to depend on complete capsid assembly (Fig. 3 and Extended Data Figs. 3, 4 and 5). This confirms that the HIV-1 capsid can generally traverse condensed FG phases and, in this context, is not adapted to a specific FG repeat sequence.

## Complete NPC passage by capsid spheres

In a next step, we injected the 40 nm capsid spheres into the cytoplasm of mouse oocytes[53] and observed not only a prominent binding to the lobulated NE but also nuclear entry and accumulation at intranuclear foci within 25 min (Extended Data Fig. 6). The effect was specific, as a much smaller, co-injected tetrameric tCherry remained entirely cytoplasmic. Thus, the 40 nm capsid spheres can traverse NPCs completely in this cell-based assay.

## Coincidence of FG binding and NPC passage defects

The CA-N57A mutation is deleterious for HIV-1 infection[54] and for binding of CA hexamers to FG motifs of Nup153 and CPSF6 (refs. 36,40;

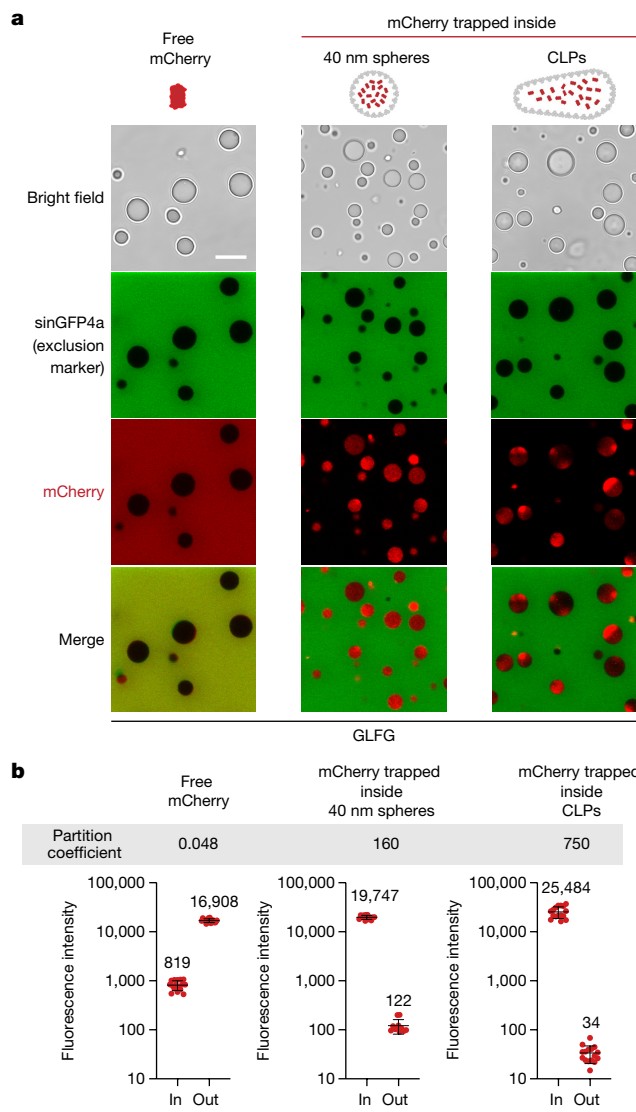

**Fig. 5 | CLPs act as cargo containers, partitioning an encapsulated client into a dense FG phase. a**, A GLFG phase (as in Fig. 3) was probed for entry of free mCherry or of mCherry that had been encapsulated in 40 nm capsid spheres or in much larger (about 60–80 nm × 100–180 nm) CLPs (Fig. 1a). Scan settings were adjusted individually. Experiments were repeated independently with the same conclusion (*n* = 5). **b**, Quantification of **a** as in Fig. 3c,d. Note that CLP-encapsulated mCherry had an at least 10,000-fold higher partition coefficient than the firmly excluded free (non-encapsulated) mCherry. Numbers are means; bars indicate mean ± s.d. *N* = number of quantified FG particles and outside areas, respectively. Scale bar, 10 μm.

Extended Data Fig. 2a). The interaction of mutant CA hexamers with FG repeats of Nup62 and Nup58 was also reduced (Fig. 2c). The mutation diminished, but did not abolish, the targeting of 40 nm capsid spheres to NPCs of HeLa cells (Fig. 4a). Likewise, the mutant capsid still bound from the cytoplasmic side to the NE/NPCs of mouse oocytes (Fig. 4c). Together this indicates that the capsids can contact FG domains in several ways.

However, the mutation severely reduced the Nup98 FG interaction, as measured by BLI (Fig. 2c), as well as the partitioning of the capsid spheres into the condensed FG phase in vitro system, with only distinct surface binding remaining (Fig. 4b). And, strikingly, it blocked the passage of cytoplasmically injected capsid spheres through NPCs into oocyte nuclei (Fig. 4c and Extended Data Fig. 7). This confirms that the partitioning of a mobile species into an Nup98 FG phase (which forms

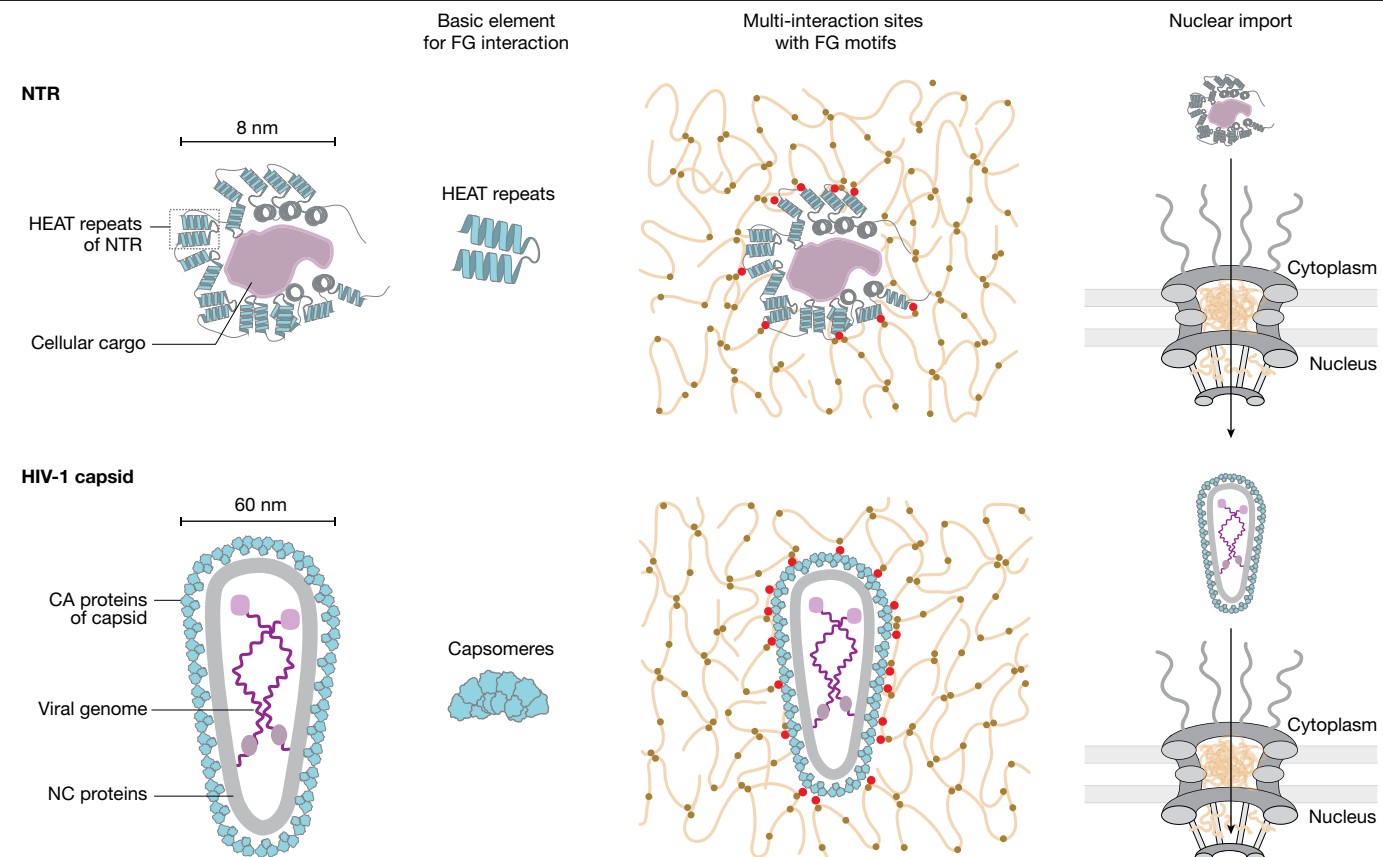

NTR

8 nm

HEAT repeats
of NTR

Cellular cargo

Basic element
for FG interaction

HEAT repeats

Multi-interaction sites
with FG motifs

Nuclear import

Cytoplasm

Nucleus

HIV-1 capsid

60 nm

CA proteins
of capsid

Viral genome

NC proteins

Capsomeres

Cytoplasm

Nucleus

**Fig. 6 | Illustration comparing an importin β-type NTR with an HIV-1 capsid.** A typical importin β-type NTR is about 8 nm in diameter and is composed of HEAT repeats that not only capture cargo but also mediate multivalent FG interactions and passage through the FG phase of NPCs. The HIV-1 capsid is composed of pentameric and hexameric capsomeres. With a diameter of 60 nm, it would be too large to pass through the NPC scaffold in an NTR-chaperoned manner. Instead, it engages directly with FG contacts, 'melting' into the FG phase and allowing the encapsulated viral genome to cross the barrier without an exposure to the cytoplasmic antiviral surveillance machinery. Cohesive inter-FG interactions are shown as brown dots, NTR/capsid–FG interactions are shown in red.

a stricter barrier than other FG domains of animal NPCs[52]) is a valid predictor for facilitated NPC passage. Taken together, these experiments suggest that the NTR-typical direct FG interactions of the capsid are not just a coincidence but a requirement for its NPC passage and for the overall infection process.

## A cargo container for enclosed clients

At this point, we wondered whether the assembled capsid would remain completely intact when entering the FG phase. As a stringent test, we assembled capsid spheres in the presence of a high concentration of non-fused mCherry. This way, mCherry molecules are encapsulated in the capsid spheres. Free mCherry was removed by a gel filtration step (Extended Data Fig. 8). In this setup, mCherry remains inside the capsid only as long as the capsid stays intact. Again, it was striking that the capsid-encapsulated mCherry entered the phase efficiently and reached a rather uniform intraphase distribution, whereas free sinGFP4a, added as an internal control to the same sample, remained phase-excluded (Fig. 5a,b). Thus, the HIV-1 capsid behaves like an NTR which enters and traverses an otherwise strictly selective FG phase. The interior of the capsid provides space for transported clients—in our 'engineered' case an inert fluorescent protein, in the case of lentiviral infection a payload of genomic RNA. Moreover, the capsid spheres seem to be thermodynamically and kinetically stable in the Nup98-like GLFG phase.

Up to this point, we have studied 40 nm capsid spheres assembled from a CA-N21C A22C double mutant[45,49]. Although this type of assembly is easy to produce, robust, disulfide-bridged and thus very stable, it differs from the actual viral capsid by its smaller size and its higher relative content of CA pentamers (with 12 pentamers and 30 hexamers[55]). By contrast, capsid-like particles (CLPs) are assembled from wild-type CA, have the typical cone shape and are similar in size to the authentic HIV-1 capsid (about 40 MDa). We produced CLPs[56,57] with encapsulated (non-fused) mCherry (Extended Data Fig. 8). Electron microscopy analysis showed them to be larger (about 60–80 nm × 100–180 nm) than the CA spheres (Fig. 1a). Nevertheless, this assembly species readily partitioned into our in vitro FG phase model system (Fig. 5), reaching a partition coefficient of more than or equal to 750—which is on par with or even higher than the partitioning of the 40 nm capsid spheres and about 15,000 times higher than non-encapsulated mCherry.

The larger size of the CLPs should result in a higher energetic barrier to phase entry but this seems to be well compensated for by the proportionally larger number of favourable FG contacts. Moreover, this experiment suggests that the FG phase per se has essentially no upper size limit for the entry of an NTR-like species.

## Discussion

Nuclear entry of the viral genome is a key step in HIV-1 infection. It occurs through NPCs of an intact NE, with the capsid apparently squeezing through the NPC scaffold[1] and the barrier-forming FG phase. Interactions between CA hexamers and specific FG motifs, in particular from CPSF6 and Nup153, have been reported and studied in depth[36–40,42]. NPC passage requires, however, the crossing of all barrier layers that

originate from about ten different FG domains. This scenario is now well supported by our observations that the capsid is a general FG repeat binder (accepting all principal FG motifs) and FG phase traverser (Figs. 1–5 and Extended Data Figs. 2–5). It is further supported by capsid spheres binding efficiently and directly to human NPCs (Fig. 1) and rapidly crossing the NE and entering nuclei following injection into the cytoplasm of mouse oocytes (Fig. 4 and Extended Data Fig. 6). Also, it is consistent with the earlier evidence supporting capsid uncoating as a nuclear event[32,33].

Despite all these data, it is important to emphasize that several aspects of HIV-1 capsid passage remain to be understood. Here, we used CLPs and capsid spheres, not bona fide (RNA-filled) HIV-1 capsids. Could the peculiar cone-shaped structure of the real HIV-1 capsid be important for NPC transport? For example, does it matter whether the HIV-1 capsid enters with its pointy or wide end or is it arbitrary? Furthermore, our FG phase model systems use rather homogeneous cohesive FG repeats, whereas real NPCs harbour a much more complex mixture of many different FG repeats which mutually interact in a highly complex manner that is far from being fully understood. Thus, several transport aspects await further investigation.

The elementary interactions between an FG repeat and an NTR are extremely weak and transient. They are amplified to physiological relevance by avidity effects resulting from the many (about 10–50) repeat units per FG domain and the several FG contact sites present in NTRs[26,58,59]. An HIV-1 capsid can be expected to contain hundreds of FG-binding sites[38–40]. Immersing the capsid into a phase containing up to 400 mg ml$^{-1}$ of FG repeats will maximize the FG–capsid contacts (Fig. 6). Consequently, the capsid should be drawn into the central NPC channel until it reaches a resting position with maximized FG interactions. Release of the capsid from this energy well and thus the completion of the NPC passage, would be slow, unless energy is supplied to the process. In contrast to import by importins, there is no obvious coupling to the RanGTPase system. However, binding of any nuclear factor that competes with capsid–FG interactions would have the same effect. CPSF6 is an excellent candidate for such competition[60–62].

Our findings add to the list of functions of the HIV-1 capsid during infection[63]. Not only does it shield the viral genome from cytoplasmic surveillance by innate antiviral factors, it also targets its concealed payload to NPCs, allowing it to cross the permeability barrier of NPC. For directed nuclear import, cellular cargoes rely on dedicated NTRs that mediate FG interactions and thus NPC passage; after cargo delivery, they return to the cytoplasm for another round of import. Why does the HIV-1 capsid deviate from this scheme and act as its own NTR? We see two explanations. First, there is no need for multiround transport with a sustained recycling reaction, given that a single successful nuclear import event of the viral genome can already establish an infection. Second, the bottleneck may be the size limit of NPCs, which can be seen as a cellular safeguard against large viral invaders. This limit is set by the NPC scaffold structure to an approximately 60 nm wide opening at its narrowest point. Passage of the 60 nm HIV-1 capsid cone might already enforce structural changes to the scaffold. However, a coating of the capsid with a layer of NTR molecules would add another 10 nm or so to the diameter and probably sterically prohibit NPC passage. Bypassing the NTR requirement by direct FG–capsid interactions thus seems to be a tailored strategy of the virus to undermine the NPC size limit, which is otherwise an effective cellular antiviral defence line.

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

## Methods

### Recombinant protein expression

Supplementary Table 1 lists plasmids used for recombinant expression. Unless noted otherwise, expression and purification were as detailed in the references (this study, and refs. 21,25,26,50,64).

### Assembly and purification of HIV-1 CA hexamers

Disulfide-stabilized CA hexamers were prepared as described in ref. 49. An amino-terminal His[14] tag for Ni(II) chelate affinity purification and a *Brachypodium distachyon* (bd)SUMO tag[65] were added for solubility to the CA[P1A/A14C/E45C/W184A/M185A] mutant. Recombinant expression was in *Escherichia coli* LOBSTR-RIL(DE3) (Kerafast) cells[66]. Liquid cultures were induced with 0.5 mM IPTG at 18 °C for 16 h. Cells were collected by centrifugation, resuspended in lysis buffer (50 mM Tris/HCl pH 8.0, 500 mM NaCl, 20 mM imidazole, 1 mM TCEP, 2 mM PMSF) and lysed using a high-pressure cell homogenizer (Microfluidics LM20). The lysate was cleared by centrifugation at 8,500$g$ for 30 min. The soluble fraction was incubated with Ni Sepharose 6 Fast Flow beads (GE Healthcare) for 1 h at 4 °C. The beads were subsequently washed with wash buffer (50 mM Tris/HCl pH 8.0, 40 mM imidazole, 300 mM NaCl, 1 mM TCEP). Proteins were eluted from the beads by tag cleavage with 2–5 µg ml⁻¹ of bdSenP1 in cleavage buffer (50 mM Tris/HCl pH 8.0, 150 mM NaCl, 0.5 mM TCEP) for 3 h at 4 °C.

Assembly was performed in a high-salt buffer containing 50 mM Tris/HCl pH 8.0, 1 M NaCl, 100 µM IP6 (myo-inositol hexaphosphate) for 24 h at a protein concentration of 10–15 mg ml⁻¹ at room temperature. The sample was then dialysed against 50 mM Tris/HCl pH 8.0. Hexamers were isolated by size exclusion chromatography (SEC) using a Superdex 200 increase column (Cytiva) in 20 mM Tris/HCl pH 8.0, 150 mM NaCl, 100 µM IP6. Samples were flash-frozen in liquid nitrogen and stored at −80 °C for later use. Protein purity was validated by SDS–polyacrylamide gel electrophoresis.

### Purification and assembly of 40 nm capsid spheres

For assembly of 40 nm capsid spheres, His[14]-bdSUMO-tagged CA[P1A/N21C/A22C] was purified as described above and concentrated to approximately 15 mg ml⁻¹. Assembly was performed in 50 mM Tris/HCl pH 8.0, 1 M NaCl and 0.1 mM IP6 for 2–48 h. The 40 nm capsid spheres were concentrated and isolated from unassembled CA by ultracentrifugation at 280,000$g$ for 2.5 h. The pellet was washed and resuspended in assembly buffer. Capsid spheres were further purified by SEC on a Superose 6 10/300 increase column equilibrated in 50 mM Tris/HCl pH 8.0, 500 mM NaCl and 0.5 mM IP6. Assembly was performed in the presence of 2 mM soluble mCherry or with the addition of CA–sinGFP4a or CA–EGFP fused tracers (at one-sixth of the molar concentration of the non-fused CA).

### Purification and assembly of CLPs

For assembly of CLPs, untagged CA[wt] was overproduced in *E. coli* LOBSTR-RIL(DE3) cells (Kerafast). Cells were collected by centrifugation and resuspended in lysis buffer (50 mM Tris/HCl pH 8.0, 50 mM NaCl, 4 mM TCEP, 2 mM PMSF and Turbo Nuclease). Cells were lysed and cleared by centrifugation at 20,000$g$ for 60 min. CA was precipitated from the cleared lysate with ammonium sulfate at a final concentration of 25% saturation for 30 min. Precipitated protein was collected by centrifugation at 9,000$g$ for 20 min. The supernatant was discarded and the pellet resuspended in lysis buffer and dialysed overnight against 20 mM Tris/HCl pH 8.0 and 1 mM TCEP. CA was further purified by injecting on a HiTrapQ column and eluting it with 20 mM Tris/HCl pH 8.0, 100 mM NaCl, 1 mM TCEP. The column was regenerated with 50 mM Tris/HCl pH 8.0, 2.0 M NaCl, 1 mM TCEP to remove nucleic acid contaminations. The eluted CA was buffer exchanged by SEC (Superdex 200 10/300 increase) into 25 mM Tris/HCl pH 8.0. Fractions were pooled and concentrated to approximately 19.0 mg ml⁻¹.

CLPs containing mCherry cargo were prepared in the presence of 650 µM of mCherry in the assembly reaction and by warming the protein mixture to 37 °C for initially 5 min. Assembly was induced by bringing the final assembly mixture to 25 mM Tris/HCl pH 8.0, 50 mM NaCl, 5 mM IP6 and 1 mM TCEP at a CA concentration of 12 mg ml⁻¹ for 1–2 h at 37 °C. Assembled CLPs were then transferred to ice and larger tube aggregates were removed by centrifugation at 20,000$g$ for 5 min. CLPs were further isolated from unassembled and tube particles by SEC on a Superose 6 10/300 increase column in 25 mM Tris/HCl pH 8.0, 100 mM NaCl, 1 mM IP6 and 1 mM TCEP.

### Negative-stain electron microscopy

A total of 4 µl of purified CA hexamer, CA-N21C/A22C capsid spheres and CLPs at 0.025, 0.1 and 0.25 mg ml⁻¹, respectively, were adsorbed for 1 min on carbon-coated 300-mesh copper grids (EMS) glow-discharged for 1 min at −15 mA using a PELCO easiGlow instrument. Sample droplets were blotted with filter paper held perpendicular to the grids. The grids were then quickly washed three times in 25 µl MilliQ water droplets, followed by staining in 25 µl droplets of 2% uranyl acetate solution for 10 s and again for 1 min, with blotting between droplets. After the final staining, grids were thoroughly blotted with filter paper and allowed to air dry before imaging. Micrographs were collected on a FEI Tecnai G2 Spirit TWIN 120 kV transmission electron microscope equipped with a Gatan Ultrascan 2k × 2k CCD detector. Representative micrographs were imaged with a defocus range of −1 to −2 µm at nominal magnifications of ×52,000 for CA hexamers or ×15,000 for capsid spheres and CLPs with pixel sizes of 4.15 and 1.25 Å, respectively.

### Digitonin-permeabilized cell assays

HeLa-K cells were obtained from the European Cell Culture Collection (RRID:CVCL_1922), authenticated by the manufacturer and tested negative for mycoplasma. Cells were grown in Dulbecco's Modified Eagle's Medium (DMEM, high-glucose), supplemented with fetal calf serum and antibiotics (AAS, Sigma-Aldrich) on 8-well µ-slides (IBIDI) to 70% confluence. Plasma membranes were permeabilized by treating the cells with 30 µg ml⁻¹ of digitonin (water-soluble fraction) in transport buffer (20 mM HEPES/KOH pH 7.5, 110 mM potassium acetate, 5 mM magnesium acetate, 0.5 mM EGTA, 250 mM sucrose) for 3 min at 25 °C (with gentle shaking), followed by three washing steps in transport buffer. Permeabilized cells were then incubated for 30 min with 40 nM Alexa647-labelled anti-Nup133 nanobodies[47] and EGFP (3 µM), CA–EGFP (0.2 µM), 40 nm viral capsid spheres (2.8 nM) or a complex formed with 200 nM IBB–GFP and 400 nM importin β. Where indicated, 3 µM of a RanQ69L fragment comprising residues 1–180 and charged with GTP was also added. The samples were then directly scanned with a Leica SP8 confocal laser-scanning microscope (equipped with a ×63 oil objective and HyD GaAsP detectors), using the 488 and 638 nm laser lines sequentially for excitation. Such live scan directly reads the distributions of the analysed probes between bulk buffer and NPCs—although local concentration at NPCs is underestimated because of the diffraction-limited resolution.

### Expression and purification of peptides for biolayer interferometry

Peptides were expressed recombinantly in *E. coli* LOBSTR-RIL(DE3) (Kerafast) cells. Cultures were induced with 0.2 mM IPTG and grown at 18 °C overnight. Cells were collected through centrifugation at 6,000$g$, resuspended in lysis buffer (50 mM Tris/HCl pH 8.0, 500 mM NaCl, 30 mM imidazole). Cells were lysed through a microfluidizer at 18,000 psi. Then 0.2 mM PMSF and Turbo Nuclease were added to lysate. Lysates were spun at 8,500$g$ for 30 min. The soluble fraction was combined with Ni Sepharose 6 Fast Flow beads (Cytiva) and incubated for 1 h at 4 °C with gentle agitation. Samples were spun to collect Ni-NTA protein bound beads. Beads and bound protein were washed with wash buffer (50 mM Tris/HCl pH 8.0, 300 mM NaCl, 40 mM imidazole).

Beads were washed with 6 column volumes of elution buffer (50 mM Tris/HCl pH 7.4, 300 mM NaCl, 250 mM imidazole). The eluted protein was dialysed into buffer (50 mM Tris/HCl pH 7.5, 150 mM NaCl). Samples were flash-frozen with liquid nitrogen and stored at −80 °C for later use. Purified proteins were validated by SDS–PAGE.

### BLI assays and analyses

High-precision streptavidin biosensor tips were pre-incubated for 10 min in BLI buffer (20 mM Tris/HCl pH 8.0, 300 mM NaCl, 0.1% (w/v) bovine serum albumin, 0.02% (v/v) Tween-20). Tips were then dipped into BLI buffer for 60 s. Next, biotinylated, N-terminally Avi-tagged ligands prepared in BLI buffer solution were immobilized to a thickness of 1 nm for 30–45 s. Ligand-loaded tips were then dipped into wells containing BLI buffer for 60 s, after which they were moved into a solution of HIV-1 CA hexamer in BLI buffer. Association was recorded for 180–300 s, followed by a 200–300 s dissociation step in wells containing BLI buffer. All binding sensorgrams were recorded on a forteBIO OctetRED96 instrument. A reference sensor was included and used to subtract background noise. Data were normalized to the baseline step and aligned to the dissociation start point using the Octet data analysis software. Data were plotted using PRISM.

### Microinjections

Mouse oocytes were obtained from ovaries of 9-week-old CD1 mice which were maintained in a specific pathogen-free environment according to the Federation of European Laboratory Animal Science Association guidelines and recommendations, as previously described[53]. Fully grown oocytes were kept arrested in prophase in homemade M2 medium supplemented with 250 μM dibutyryl cyclic AMP under paraffin oil (NidaCon) at 37 °C. Labelled 40 nm capsid spheres (along with the tetrameric Cherry injection marker Sin_tCherry2; ref. 26) were microinjected into cytosol or nucleus of oocytes, as previously described[53]. Oocytes were imaged about 25 min after injection.

### FG phase assays

The assays were performed as previously described[25,50] with minor modifications. In brief, phase separation was initiated by rapidly diluting a 1 mM FG domain stock with 25 volumes (for GLFG) or 50 volumes (for the others) of assay buffer (50 mM Tris/HCl pH 7.5, 250 mM NaCl, 1 mM IP6), followed by another fourfold dilution in buffer containing the indicated fluorescent probes. The resulting mixture was placed on collagen-coated μ-slides 18-well (IBIDI) and FG particles were allowed to sediment to the bottom of the slides for 1 h before confocal scans were taken.

The GLFG stock was dissolved in 2 M guanidinium hydrochloride and the other FG stocks in 4 M guanidinium hydrochloride. Assay buffer for human Nup98-FGs and yeast Nup116-FGs was 50 mM Tris/HCl pH 7.5, 150 mM NaCl, 1 mM IP6.

Partition coefficients were calculated as the raw signal in independent FG particles (in) divided by the reference areas in outside regions (out). Plots are shown for representative FG particles (with 4–7 μm diameters). Images were analysed in FIJI 2.9.0 and the exported data were further processed in GraphPad Prism 9.5.1.

The sequences of FG domains used in FG phase assays are shown in Supplementary Fig. 1.

### Reporting summary

Further information on research design is available in the Nature Portfolio Reporting Summary linked to this article.

## Data availability

Data supporting the findings of this study are available in the article and its Extended Data and Supplementary Information. Source data are provided with this paper.

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

**Acknowledgements** We thank S.C. Ng, J. Schünemann and H.B. Schmidt for providing Nup98 FG repeat domains for phase-separation experiments, K. Gregor for the labelled anti-Nup133 nanobody and M. Schuh for generous support of the oocyte injection experiment. We also thank D. Riedel and L. Kopecny of the Electron Microscopy Facility, A.Z. Politi of the Light Microscopy Facility and the staff of the Animal Facility of the Max Planck Institute for Multidisciplinary Sciences for service and technical assistance. Negative-stain EM was carried out in the MIT. nano facility. We further thank O. Pornillos and B. Pornillos for invaluable technical advice and A. Engelman and J. Luban for critical discussions. We thank D. Jacques and his team for sharing their capsid-NTR data before publication. This project received funding from the Deutsche Forschungsgemeinschaft (SFB 1190). This work was also supported by NIH grant nos R35GM141834 and R21AI179432 (to T.U.S.). NIH training grant no. T32GM007287 supported E.N.W.

**Author contributions** D.G. and T.U.S. conceived and oversaw the project and interpreted experimental results with contributions from all other authors. D.G. and L.F. designed and L.F. performed and interpreted the permeabilized cell and FG-phase assays. E.N.W. designed, performed and interpreted the BLI experiments, with initial input from O.A. O.A. purified proteins and designed reagents (including mCherry trapped capsids). N.A.S. performed EM analysis. L.F. and S.C. designed and performed the oocyte injection experiments. D.G. and T.U.S. wrote the manuscript with contributions from E.N.W. All authors edited and approved the manuscript.

**Funding** Open access funding provided by Max Planck Society.

**Competing interests** The authors declare no competing interests.

**Additional information**
**Correspondence and requests for materials** should be addressed to Thomas U. Schwartz or Dirk Görlich.

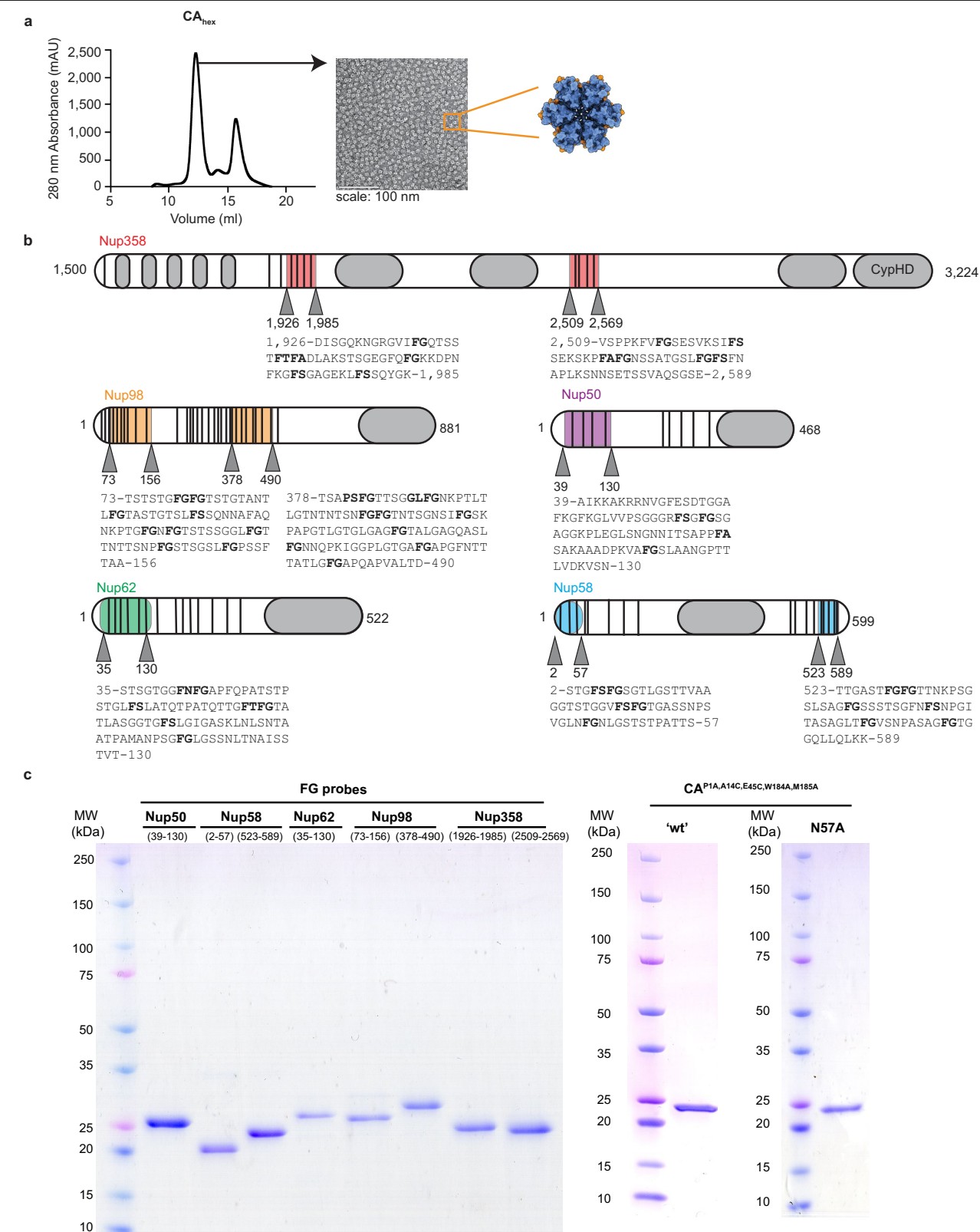

**Extended Data Fig. 1 | BLI constructs. a**, Purification of disulfide-stabilized CA hexamers (CA-P1A, A14C, E45C, W184A, M185A). Gel-filtration profile from a Superdex 200 (HR10/300) column with CA hexamers eluting as the major peak. The peak fraction was analysed by negative-stain EM. **b**, Illustration of the FG peptides used as BLI-probes in Fig. 2. The illustrations are drawn to scale.

Vertical lines indicate individual FG- and FG-like dipeptides (including FG/S/T/A), the coloured sections with boundaries denote the actual FG-domain fragments tested, with the amino acid sequences also shown. Structured domains are depicted in grey. CypHD, cyclophilin-homology domain. **c**, SDS–PAGE analysis of all BLI-probes used in Fig. 2.

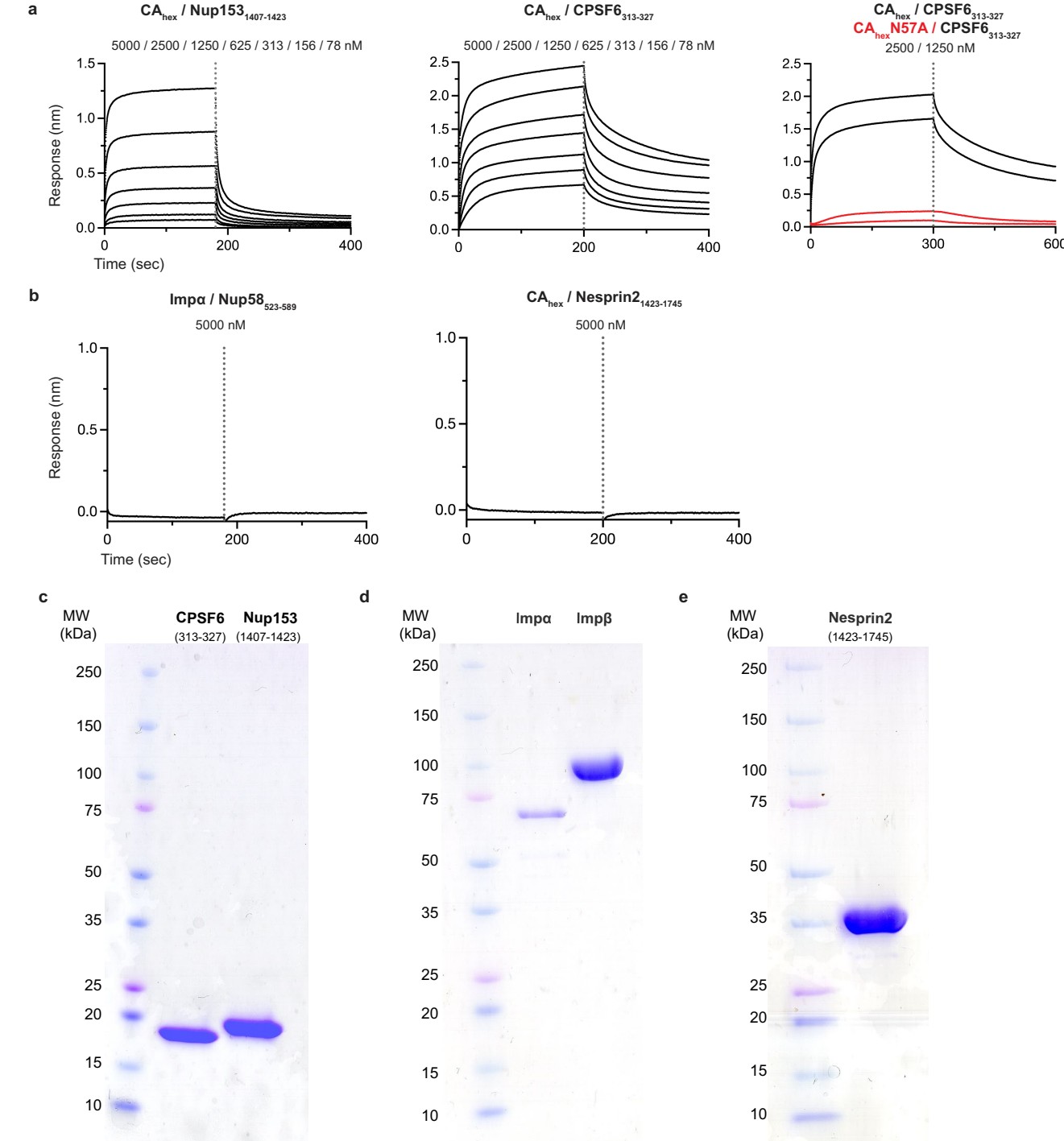

**Extended Data Fig. 2 | BLI controls. a**, Control BLI experiments showing that CA hexamer binds the established Nup153 and CPSF6 FG peptides. The CA-N57A mutant abolishes CPSF6 binding. Analyte concentrations of the dilution series are indicated. **b**, BLI-negative controls: Representative of FG probes, Nup58$_{523-589}$ does not interact non-specifically with the analyte, as shown with importin α, which is not known to interact directly with FG peptides. Conversely, CA hexamers do not interact non-specifically with unrelated ligands, as shown with a Nesprin2$_{1423-1745}$ fragment. Both negative control experiments were performed with 5 μM analyte. **c**–**e**, SDS–PAGE analysis of BLI control probes.

**a** **Z-stack** (X-Y plane, from top to bottom)

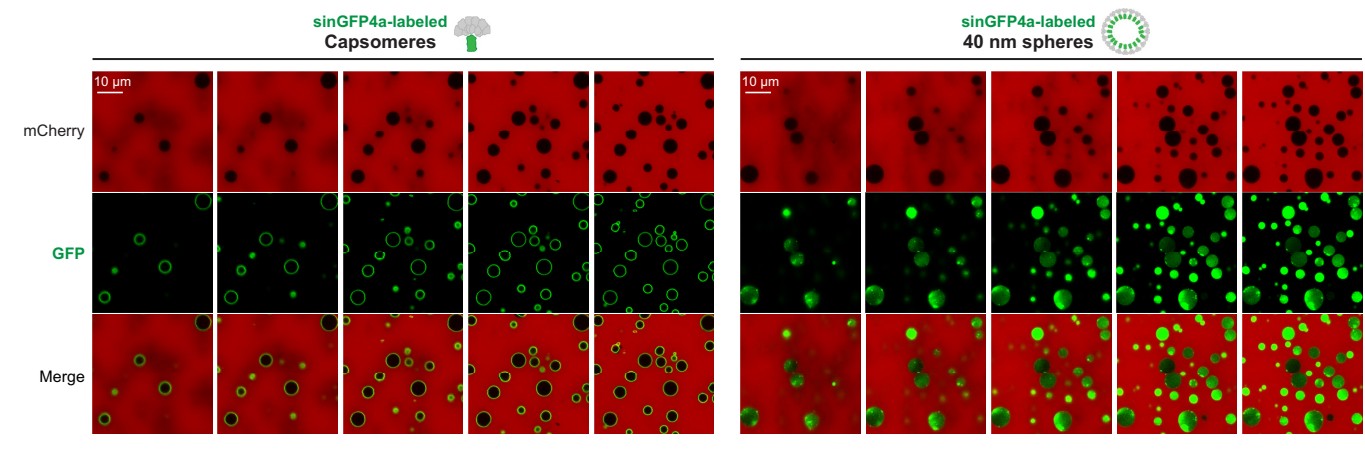

**b** **Y-stack** (X-Z plane)

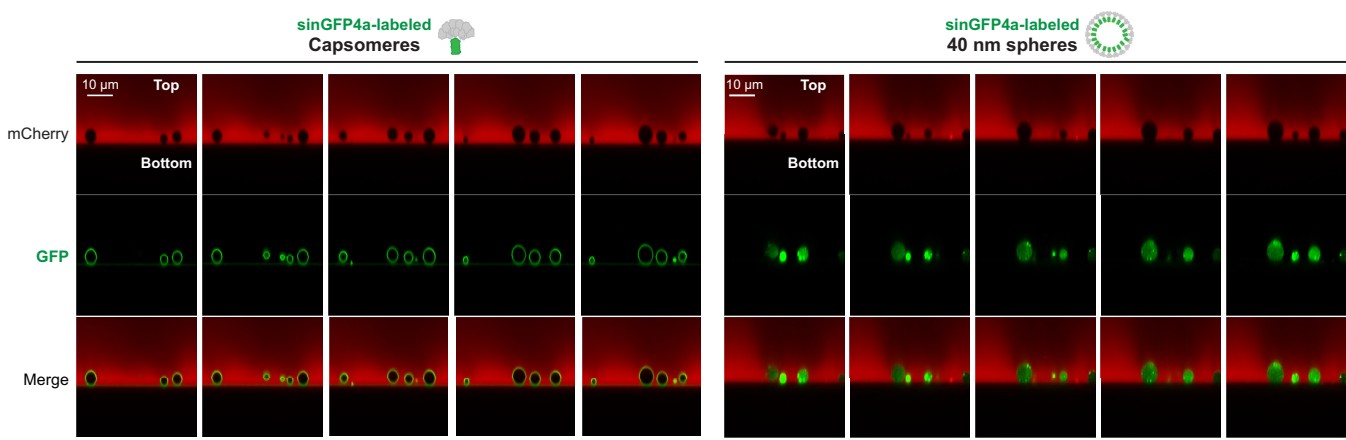

**Extended Data Fig. 3 | Representative confocal stack images showing the distribution of capsomeres and 40 nm capsid spheres in FG phases.** FG phase partitioning experiments were performed as in Fig. 3, using the sinGFP4a-fused capsomeres and 40 nm capsid spheres. Scan settings were adjusted individually. Z-stack (**a**) and Y-stack (**b**) images of FG particles with different sizes were shown.

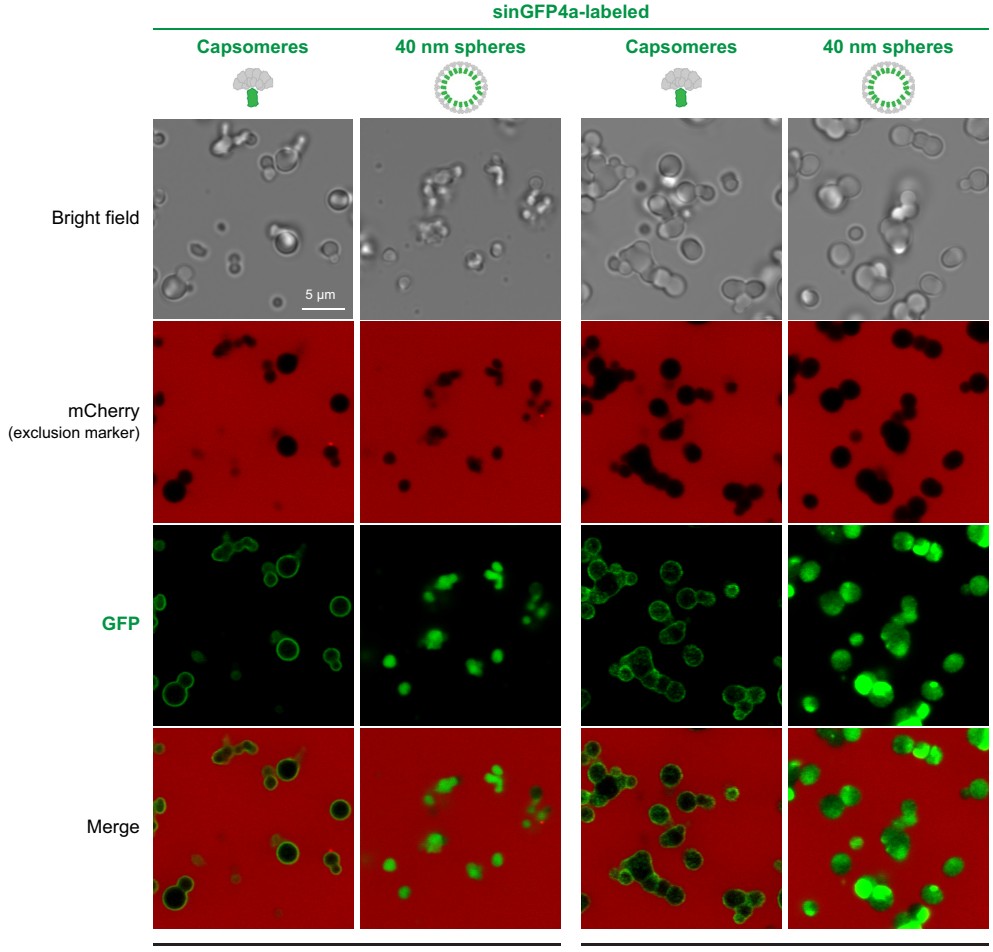

**Extended Data Fig. 4 | Partition of capsid spheres into Nup98-related FG phases from yeast and human.** FG phase partitioning experiments were performed as in Fig. 3, using the indicated FG phases and capsid species. Experiments were repeated independently with the same outcome (n = 4).

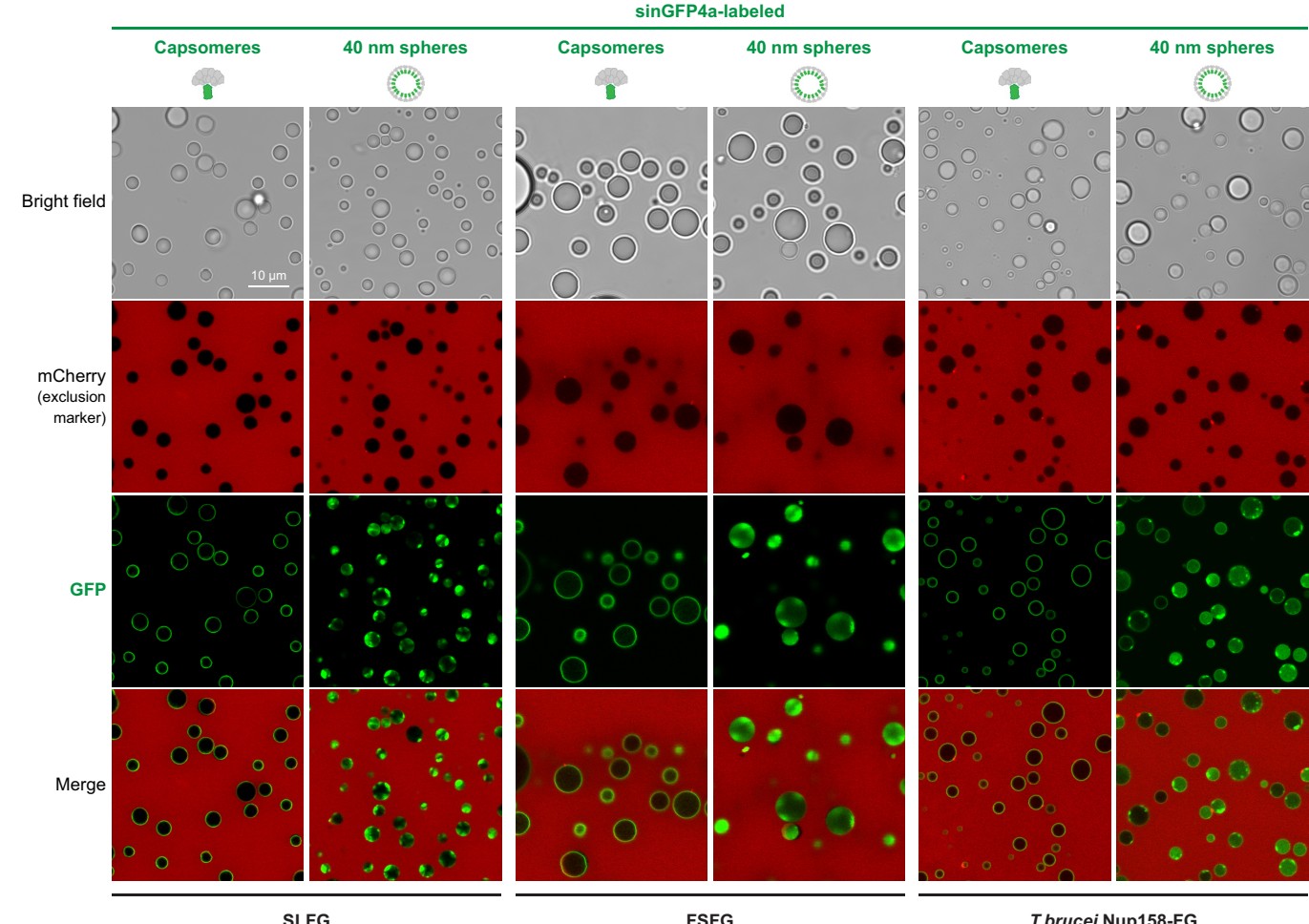

**Extended Data Fig. 5 | Capsid entry into diverse FG phases.** Experiment was as in Fig. 3e but showing in addition the controls with a partially assembled capsomere–sinGFP4a fusion. Experiments were repeated independently with the same outcome (n = 4).

**Extended Data Fig. 6 | Capsid spheres can complete NPC passage and enter the nuclei of mouse oocytes. a**, Labelled 40 nm capsid spheres were injected (along with a tetrameric tCherry injection marker) either into the cytoplasm or the nucleoplasm of fully grown mouse oocytes. Images show confocal laser scans 25 min after injection, when the endpoint distribution had already been reached. Note that the injection marker remained in the injected compartment, while cytoplasmically injected capsid spheres readily crossed the nuclear envelope and accumulated in intranuclear foci (indicated by white arrows in the zoomed-in images). The differences to Fig. 4c are (i) that complete oocytes were imaged here and not just smaller areas of interest and (ii) that the capsids were labelled with the FG-phobic sinGFP4a, which needs to be buried inside the capsid to avoid an arrest at the barrier (see Fig. 3b,d). Experiments were repeated independently with the same outcome (n = 3). **b**, Scheme of a fully grown mouse oocyte.

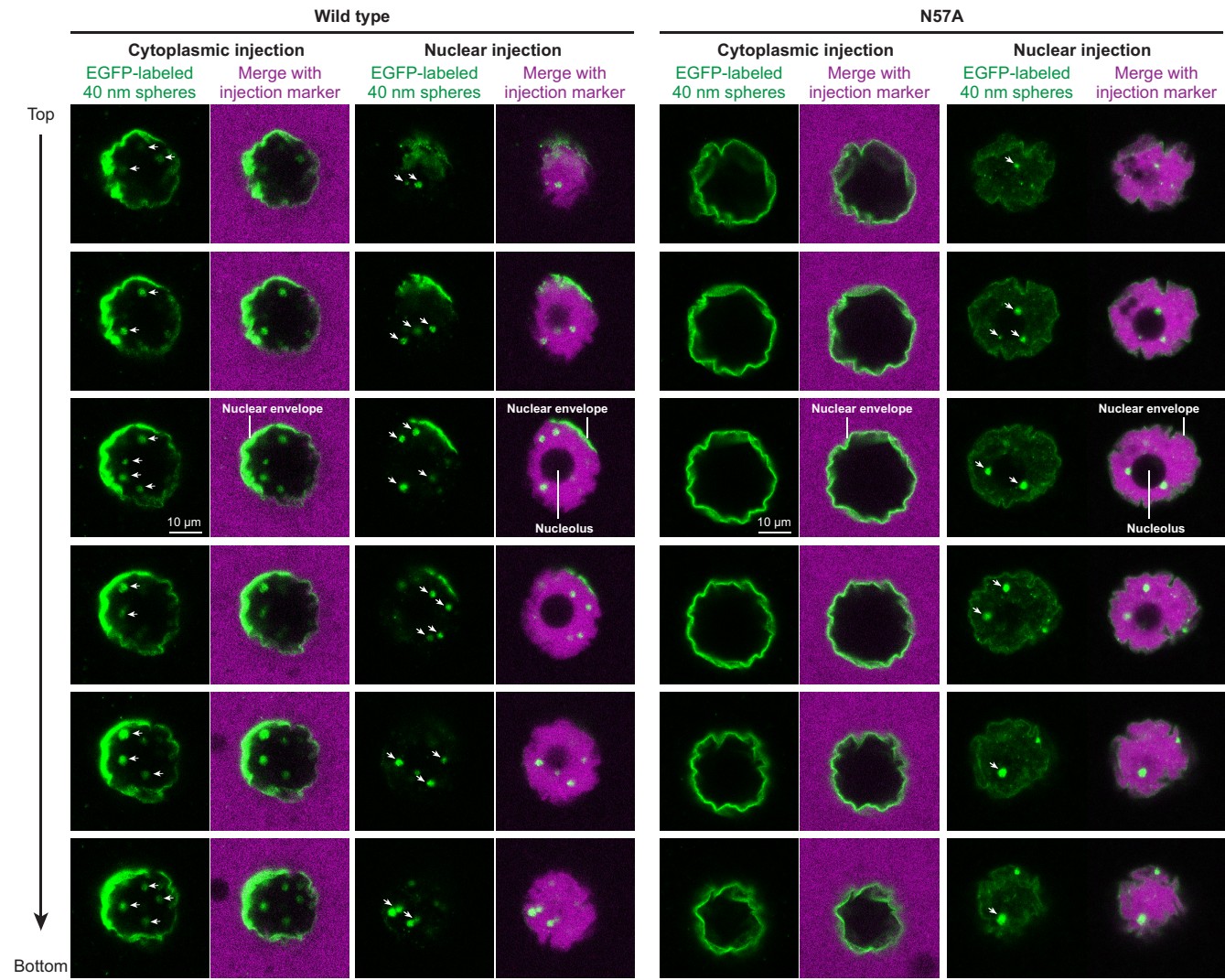

**Extended Data Fig. 7 | Capsid spheres with N57A mutation fail to complete NPC passage.** Z-stacks of the samples shown in Fig. 4c.

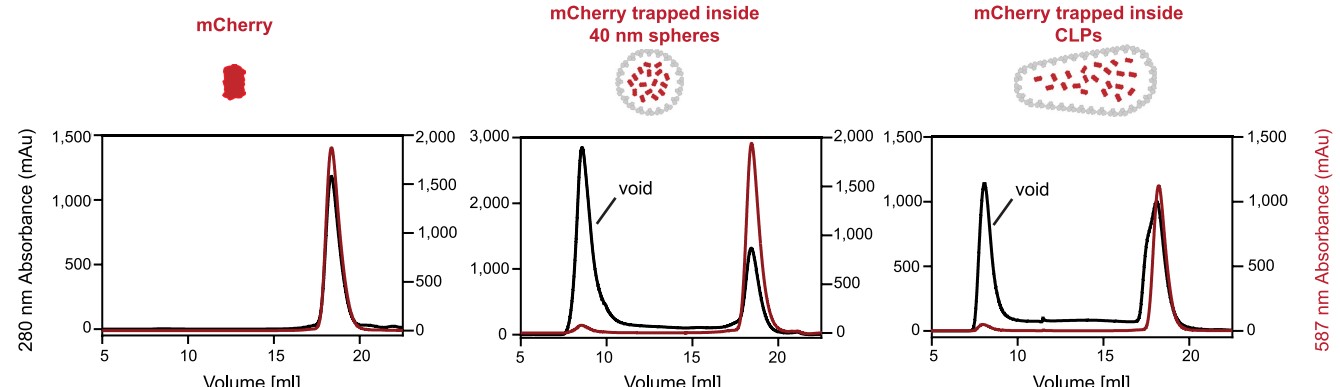

**Extended Data Fig. 8 | Preparation of 40 nm capsid spheres and CLPs with mCherry trapped inside.** After the assembly of 40 nm capsid spheres and CLPs in the presence of excess amounts of mCherry, the particles were separated from unincorporated mCherry by gel filtration (Superose 6 10/300 column). Absorbance traces were recorded at 280 nm (total protein) and 587 nm (mCherry). **a**, Control experiment with free mCherry, showing no detectable aggregation (no void peak). **b**, Assembled 40 nm capsid spheres elute in the void volume, together with a fraction of the mCherry, indicating encapsulation. **c**, Assembled CLPs show a behaviour similar to **b**, with the encapsulated mCherry fraction co-eluting in the void peak.

# Reporting Summary

## Statistics

For all statistical analyses, confirm that the following items are present in the figure legend, table legend, main text, or Methods section.

| n/a | Confirmed | |
|---|---|---|
| ☐ | ☒ | The exact sample size (*n*) for each experimental group/condition, given as a discrete number and unit of measurement |
| ☒ | ☐ | A statement on whether measurements were taken from distinct samples or whether the same sample was measured repeatedly |
| ☒ | ☐ | The statistical test(s) used AND whether they are one- or two-sided *Only common tests should be described solely by name; describe more complex techniques in the Methods section.* |
| ☒ | ☐ | A description of all covariates tested |
| ☒ | ☐ | A description of any assumptions or corrections, such as tests of normality and adjustment for multiple comparisons |
| ☐ | ☒ | A full description of the statistical parameters including central tendency (e.g. means) or other basic estimates (e.g. regression coefficient) AND variation (e.g. standard deviation) or associated estimates of uncertainty (e.g. confidence intervals) |
| ☒ | ☐ | For null hypothesis testing, the test statistic (e.g. *F*, *t*, *r*) with confidence intervals, effect sizes, degrees of freedom and *P* value noted *Give P values as exact values whenever suitable.* |
| ☒ | ☐ | For Bayesian analysis, information on the choice of priors and Markov chain Monte Carlo settings |
| ☒ | ☐ | For hierarchical and complex designs, identification of the appropriate level for tests and full reporting of outcomes |
| ☒ | ☐ | Estimates of effect sizes (e.g. Cohen's *d*, Pearson's *r*), indicating how they were calculated |

*Our web collection on statistics for biologists contains articles on many of the points above.*

## Software and code

Policy information about availability of computer code

| Data collection | was using commercial instruments and the included software (FEI Tecnai G2 Spirit TWIN 120 kV transmission electron microscope, Leica SP8 confocal laser scanning microscope, Zeiss LSM 880 confocal laser scanning microscopes, forteBIO OctetRED96) |
|---|---|
| Data analysis | Instrumental software, FIJI 2.9.0 (Schindelin et al., 2012 Fiji: an open-source platform for biological-image analysis. Nat Methods 9, 676-682) and GraphPad Prism 9.5.1 (GraphPad Software, Boston, Massachusetts USA, www.graphpad.com) |

For manuscripts utilizing custom algorithms or software that are central to the research but not yet described in published literature, software must be made available to editors and reviewers. We strongly encourage code deposition in a community repository (e.g. GitHub). See the Nature Portfolio guidelines for submitting code & software for further information.

## Data

Policy information about availability of data

All manuscripts must include a data availability statement. This statement should provide the following information, where applicable:
- Accession codes, unique identifiers, or web links for publicly available datasets
- A description of any restrictions on data availability
- For clinical datasets or third party data, please ensure that the statement adheres to our policy

All data that are described in the manuscript are also explicitly shown in the figures

April 2023

# Research involving human participants, their data, or biological material

Policy information about studies with human participants or human data. See also policy information about sex, gender (identity/presentation), and sexual orientation and race, ethnicity and racism.

| | |
|---|---|
| Reporting on sex and gender | not applicable |
| Reporting on race, ethnicity, or other socially relevant groupings | not applicable |
| Population characteristics | not applicable |
| Recruitment | not applicable |
| Ethics oversight | not applicable |

Note that full information on the approval of the study protocol must also be provided in the manuscript.

# Field-specific reporting

Please select the one below that is the best fit for your research. If you are not sure, read the appropriate sections before making your selection.

☒ Life sciences ☐ Behavioural & social sciences ☐ Ecological, evolutionary & environmental sciences

For a reference copy of the document with all sections, see nature.com/documents/nr-reporting-summary-flat.pdf

# Life sciences study design

All studies must disclose on these points even when the disclosure is negative.

| | |
|---|---|
| Sample size | No statistical methods were used to predetermine sample size. The effects described here are obvious, reproducible, clear, and homogeneous enough that no statistical methods were required to extract the trait. For example, all inspected cells on a coverslip showed NPC-targeting of capsids, and all performed FG-phase experiments showed extremely strong intra-phase capsid accumulation (to a partition coefficient of ≥100 which is >1000 times higher than GFP or mCherry alone). For partition coefficient calcualtion, FG particles numbers >10 is enough to cover the effects. |
| Data exclusions | None. |
| Replication | Seven times for Figure 1b, three times for Figure 1c and 1d; four times for Figure 3a and 3e, thirteen times for Figure 3b; three times for Figure 4; five times for Figure 5. Four times for Extended Data Figure 3 and 4, three times for Extended Data Figure 5. All attempts for replication were successful. |
| Randomization | The allocations of FG particles/HeLa-Kyoto cells into wells of plates for test were random. Mouse oocytes were randomly split into different groups. |
| Blinding | Investigators were not blinded to allocation during the experiments and analysis, as each experiment was conducted by a single investigator. |

# Reporting for specific materials, systems and methods

We require information from authors about some types of materials, experimental systems and methods used in many studies. Here, indicate whether each material, system or method listed is relevant to your study. If you are not sure if a list item applies to your research, read the appropriate section before selecting a response.

## Materials & experimental systems

| n/a | Involved in the study |
|---|---|
| ☐ | ☒ Antibodies |
| ☐ | ☒ Eukaryotic cell lines |
| ☒ | ☐ Palaeontology and archaeology |
| ☒ | ☐ Animals and other organisms |
| ☒ | ☐ Clinical data |
| ☒ | ☐ Dual use research of concern |
| ☒ | ☐ Plants |

## Methods

| n/a | Involved in the study |
|---|---|
| ☒ | ☐ ChIP-seq |
| ☒ | ☐ Flow cytometry |
| ☒ | ☐ MRI-based neuroimaging |

## Antibodies

| | |
|---|---|
| Antibodies used | anti-Nup133 nanobody as described in: Colom MS, Fu Z, Güttler T, Trakhanov S, Srinivasan V, Gregor K, Pleiner T, Görlich D (2023) |
| Validation | We (the Görlich lab) are the original source and the publication cited above includes the description and in-depth-validation. |

## Eukaryotic cell lines

Policy information about cell lines and Sex and Gender in Research

| | |
|---|---|
| Cell line source(s) | HeLa-Kyoto |
| Authentication | Commercially availabe (ECACC;RRID:CVCL_1922) |
| Mycoplasma contamination | negative |
| Commonly misidentified lines (See ICLAC register) | no commonly misidentified cell line was used in the study, |

