## [Peer Review File · Nature]

Manuscript Title: HIV-1 capsids enter the FG phase of nuclear pores like a nuclear transport receptor

Redactions – unpublished data

Reviewer Comments & Author Rebuttals

Reviewer Reports on the Initial Version:

Referees' comments:

Referee #1 (Remarks to the Author):

This manuscript suggests that HIV-1 can traverse the nuclear envelope, without assistance from importins, by itself acting as a nuclear transport receptor. The model proposes the capsid itself binds FG repeats within the pore complex and holds cargo to be translocated, thus minimizing the size needed to traverse the nuclear envelope.

This model provides an explanation for how HIV and similar lentiviruses can more easily cross the nuclear barrier than other viruses. Using a combination of in vivo cell culture and in vitro phase separation experiments, direct interactions between capsid particles assembled by the authors and FG domains imply self-translocation of the HIV-1 capsid is necessary for infection. Authors hypothesize that HIV-1 capsid transport is more direct and efficient than other viral infection mechanisms since there is no need to breakdown the nuclear envelope, as the FG interactions themselves mediate the penetration of this barrier.

Overall, this is an important manuscript and should be published in Nature after addressing comments below:

Comments:

1. As this manuscript already recognizes, the nuclear transport machinery and its impact on different biological processes such as infection can vary greatly among cell types. Perhaps something to strengthen the argument, would be to show the “importin independence” is true of multiple cell types (i.e. post-mitotic cells).
2. Small note: in the figure legends under Figure 1 part b, the manuscript reads, “...confocal laser scans were taken directly through the life samples.” I think this should read “live samples”.

Co-submission shared comments:

1. In vivo results from Fu et al. nicely complements in vitro studies from Dickson et al.
2. Fu et al. shows the capsids entering other FG phases commented on in Dickson et al. – I think this better supports the phenomena that FG interactions allow the capsid to directly penetrate the nuclear pore, rather than only using Nup98 as the model FG domain as commented on earlier.
3. These two papers display a novel model for the infection of HIV that I think will be useful for the field to expand upon for other viruses.

Referee #2 (Remarks to the Author):

In this manuscript by Fu et al., the authors suggest that the principal constituent of the HIV capsid protein (CA) exhibits behavior similar to a karyopherin, forming multivalent, weak, and specific associations with the FG repeat domains in the NPC's permeability barrier. This direct interaction is suggested to be sufficient to overcome the permeability barrier, allowing the viral capsid to traverse the NPC and gain access to the nucleus in non-dividing human cells.

Methodologically, they first demonstrate that monomeric and oligomeric CA directly interact with various forms of FG domains in vitro, dependent on the integrity of the CA FG-binding pocket. Next, they use reconstituted FG domain condensates in vitro to show that capsid-like structures of CA effectively interact with FG elements of the condensates and can even infiltrate these condensates to distances corresponding to the length of the NPC central channel. The viral-like particle's interaction and merging with the condensates does not destabilize the CA oligomers, nor necessitate the presence of nuclear transport factors or RanGTP. The authors posit that the karyopherin-like arrangement of CA within the capsid lattice endows the capsid with self-translocating characteristics, accounting for the virus's ability to infiltrate the nucleus of non-dividing cells.

However, in comparison with the current state of the field, this study has significant shortcomings. Much is already known about how CA interacts with the NPC and how the HIV capsid traverses the NPC, and this work does not significantly advance our mechanistic understanding of this process. It's widely recognized that CA directly interacts with FG domains, as evidenced by multiple research groups and reviewed by researchers such as Kane et al., 2018 and Guedan, 2021. Additionally, structures illustrating the exact manner in which CA binds FG repeats have been published (e.g., Price et al., 2014; Stacey et al., 2023), detailing how a specific pocket attaches to the FG repeat. As such, it's predictable that altering this pocket would impede this interaction. Hence, the molecular mechanism of capsid interaction with FG repeats to enter the NPC is well-known.

CA is known to interact with various other cellular proteins that facilitate NPC interaction. It's established that the FG binding pocket interacts with several non-Nup proteins like CPSF6 and Sec24C. Furthermore, CA also interacts with the nuclear transport factor TNPO3 and the nucleoporin Nup358, not through its FG repeats but its cyclophilin-like domain. Despite acknowledging these known interactions, the authors overlook these additional host co-factors demonstrated to be involved in capsid entry to the NPC in their mechanistic models. Given the known dynamic nature of these interactions, asserting that adding them to the entire capsid would create an oversized shell unable to pass through the NPC is an unsupported assumption. There could be various arrangements and substoichiometric associations that enable passage, considering these proteins are known to aid capsid entry into the NPC. As per Guedan, 2021, "the viral core should be considered a dynamic structure that binds numerous cellular proteins on its path through the NPC," a contrasting view from that presented in this manuscript.

Key questions remaining in the field about this process are also not really addressed. These include questions on the sequence of events, possible NPC dilation or re-structuring, the roles of nuclear basket, Nup88 complex and cytoplasmic filaments, the coupling of viral entry to the NPC and uncoating, and the uncertainty of whether intact capsids completely traverse the NPC or

disassemble during the process. Zila et al., 2021 illustrated that the capsid enters the NPC narrow end first, suggesting NPC dilation to accommodate it, thereby pointing towards an orderly entry mechanism; their EM analysis also noted major remodeling of capsids upon NPC entry, corroborated by other groups' research. None of these pivotal steps are addressed in this work, indicating that the model presented here is inadequate.

Moreover, the conclusions that are presented are not fully substantiated by the data presented (below). Hence, I believe this manuscript is not suitable for publication in Nature. My other main general concerns are as follows:

1. The authors assume that FG macroscopic condensates replicate the properties and behavior of the nanoscale NPC permeability barrier, a premise that is not universally accepted. In fact, this assumption and in vitro system were recently rebutted in a publication in Nature (Yu et al., 2023) and a BioRxiv publication (Kozai et al., 2023), which collectively demonstrate that the behavior of FG domains in condensates does not accurately reflect the state and properties of native, NPC anchored FG Nups. Thus, many of the conclusions in this paper – based on this other work – could be incorrect. The authors do not address this serious issue.

2. Based on the data presented, I believe a more straightforward and adequately substantiated interpretation would be that the CA capsid lattice might facilitate the docking of the HIV capsid to the NPCs. However, the manuscript does not provide convincing proof that the HIV capsids or the smaller 40 nm capsid-like spheres fully traverse the NPC in vivo. The authors mention this idea, but they don't consider it a primary conclusion, instead favoring a notion of Kap-like trafficking through the NPC, a possibility that remains unverified. The integration into FG condensates isn't sufficient proof, and it would be beneficial to demonstrate this in an actual NPC.

3. Karyopherins form multivalent, weak, and transient interactions with the FGs in the NPC. It remains unclear from the evidence provided that this is how CA multimers in the HIV capsid operate. Thus, without further data and controls, the comparison of CA to a karyopherin, though intriguing, may not accurately depict the CA lattice's actual behavior and could potentially mislead the general reader.

4. The manuscript does not clearly state the actual duration of the experiments performed with these condensates. These are known, even by the authors' admission, to be susceptible to "aging", gelation, and formation of potentially aberrant beta-structures, a scenario unlikely to occur in a living NPC permeability barrier. How would the "aging" of the condensates influence the different experiments presented in these manuscripts?

5. It's concerning that the presence of karyopherins seems to significantly outcompete the CA protein or hexamer in its interaction with FG condensates. This situation would be more akin to what an HIV capsid would encounter in a living NPC: hundreds of copies of different types of karyopherins and other nuclear transport factors (carrying RNPs, Ran, etc), all interacting and passing through the NPC along with their cargoes and other molecules, creating a heterogeneous, complex, and dynamic nano-environment. The capsid would never encounter pure and homogeneous FG condensates. How do the authors reconcile these seemingly conflicting observations? The authors should show

the effect that the presence of excess karyopherins have in their FG condensate CA interaction assays, more accurately reflecting in vivo conditions.

6. The fact that the capsid spheres localize at the nuclear rim of digitonin-permeabilized HeLa cells suggests that they might indeed be docking to NPCs, however, electron microscopy could be used to verify if that's the case. Such an experiment would nicely demonstrate that the particles are not just attached to the NE and could potentially illustrate how those particles interact with the NPC. Are they attached to the periphery? Blocked in transit through the channel? Retained at the nuclear side? Do they show a continuum of all these behaviors? It would strongly support the authors' claim if they could show and analyze the attachment and localization of those particles in native, active NPCs, also nicely complementing their in vitro assays with an in-situ analysis.

7. The localization of the capsomeres and the CA-spheres in the FG condensates is quite heterogeneous. The capsomeres seem to accumulate at the rim of the big condensates, but in the smaller ones, they do seem to penetrate further inside (see Ext.Data Fig. 3d), showing a much thicker rim stain, is this caused by an optical effect or is it a real property dependent on the size of the condensate? In the case of the CA-spheres, there are particles in which most of the signal seems to concentrate at the rim of the condensate, and in other penetrates non-homogeneously in the condensate, even forming clear bright concentrated regions within the condensates. Could this be caused by the formation of amyloids in the condensates as they age? The condensates are presumably not homogeneous, as shown recently in the Lim lab BioRxiv paper - I would like to ask the authors to comment and explain these effects.

Referee #3 (Remarks to the Author):

Fu and colleagues study FG peptide binding to CA hexamers by BLI and from here study the ability of CLPs (spheres and cone-like) to enter into artificial FG hydrogels. The data is consistent with the interpretation that the HIV-1 capsid is likely to serve as its own NTR.

One problem with this paper is apparent mis-citing between text and figure panels. This surely complicates data interpretation for readers. This moreover gives the impression of a rush job. One is struck by a somewhat similar though more comprehensive study available on bioarchives, which possibly impacted compilation speed here. Nevertheless, this paper incorporates the unique control to incorporate a soluble fluid marker into CLPs and show that this marker stays incorporated following hydrogel intrusion.

The paper has no virology data and the authors seem to have limited virology experience. This is evident by some blatant mis-statements. The authors also cite only some of the relevant HIV literature.

1) The authors target five Nup proteins for study Ext Data Fig 2b. Have any of these proteins been shown previously to bind CA? Have knockdown cells been challenged with HIV? What are the results of these experiments? Are these data consistent with roles for these Nups in HIV nuclear import?

2) Page 3 line 2. Retrovirus genetic material is RNA. It is the DNA form made by reverse transcription that is integrated.

3) 4 lines from bottom. HIV is not endocytosed. Its membrane fuses with the cellular plasma membrane.

4) Please cite the IBB control experiment Fig 1c in main text page 4.

5) Page 5 line 1, "and CPSF6 peptide controls (Extended Data Fig 2c)"

6) Line 2, cite Fig 2c after "CA-N57A mutation"

7) Line 3, cite (Extended Data Fig 2c) at end of sentence.

8) Line 4, Fig 2c should apparently be Fig 2b.

9) 2nd full paragraph line 3, for clarity, please indicate (Fig. 3d).

10) Ending sentence "But still, ...". Where is this data shown?

11) Page 6 line 4, please add citations after "the capsid spheres."

12) Next paragraph line 4, please show separation of free mCherry by gel filtration.

13) Line 8, Fig 3b should apparently be Fig 3d?

14) 6 lines from bottom, Fig 3c and Ext Data Fig 3b should apparently be Fig 3d and Ext Data Fig 3c. Also, shouldn't at least 500 be "approximately 26,000"?

15) Page 7 middle paragraph line 2, please also cite ref 24, which first showed this for Nup153.

16) Please avoid "It is not exactly known if the capsid ever completes NPC passage". Ref 15, 16 images convincingly show incursion into the nuclear lumen before apparent HIV uncoating for integration. Moreover, viruses unable to bind CPSF6 (through knockdown or CA mutation) arrest at the NPC and integrate into novel genomic regions at the nuclear periphery. Thus, while CPSF6 deficient viruses seem perhaps unlikely to complete NPC passage, there is solid evidence to suggest otherwise under basal infection conditions.

17) Page 9 Fig 1b legend seems to omit capsomere description. Line 6, "life" should be "live".

18) Final word page 9, "equally" is gross over-statement. For Nup58, there is only 2-fold reduction, while affects onCPSF6 and Nup98 seem 10-fold or greater. Nup62 is somewhere in between.

19) Page 11 line 1, as meant? Doesn't seem that CA-mEGFP is in this figure.

20) Please avoid CypA label at Nup358 C-terminus, Ext Data Fig 2b. There are nearly 20 human cyclophilins, and CypA is a standalone protein. Consider CypH for cyclophilin homology domain.

Author Rebuttals to Initial Comments:

Referee #1 (Remarks to the Author):

This manuscript suggests that HIV-1 can traverse the nuclear envelope, without assistance from importins, by itself acting as a nuclear transport receptor. The model proposes the capsid itself binds FG repeats within the pore complex and holds cargo to be translocated, thus minimizing the size needed to traverse the nuclear envelope.

This model provides an explanation for how HIV and similar lentiviruses can more easily cross the nuclear barrier than other viruses. Using a combination of in vivo cell culture and in vitro phase separation experiments, direct interactions between capsid particles assembled by the authors and FG domains imply self-translocation of the HIV-1 capsid is necessary for infection. Authors hypothesize that HIV-1 capsid transport is more direct and efficient than other viral infection mechanisms since there is no need to breakdown the nuclear envelope, as the FG interactions themselves mediate the penetration of this barrier.

Overall, this is an important manuscript and should be published in Nature ...

Thank you!

...after addressing comments below:

Comments:

1. As this manuscript already recognizes, the nuclear transport machinery and its impact on different biological processes such as infection can vary greatly among cell types. Perhaps something to strengthen the argument, would be to show the “importin independence” is true of multiple cell types (i.e. post-mitotic cells).

We could test other cell types, but we do not expect new insights given that NPCs function very much the same in all nucleated cells. It is true, Lentiviruses are special in their ability to infect post-mitotic cells, however, we believe we have addressed this by using interphase HeLa cells, which have intact nuclear envelopes and NPCs (in mitosis, these structures would be disassembled).

2. Small note: in the figure legends under Figure 1 part b, the manuscript reads, “...confocal laser scans were taken directly through the live samples.” I think this should read “live samples”. Corrected as suggested.

Co-submission shared comments:

1. In vivo results from Fu et al. nicely complements in vitro studies from Dickson et al.

Thank you!

2. Fu et al. shows the capsids entering other FG phases commented on in Dickson et al. – I think this better supports the phenomena that FG interactions allow the capsid to directly penetrate the nuclear pore, rather than only using Nup98 as the model FG domain as commented on earlier.

Thank you!

3. These two papers display a novel model for the infection of HIV that I think will be useful for the field to expand upon for other viruses.

Thank you!

Referee #2 (Remarks to the Author):

In this manuscript by Fu et al., the authors suggest that the principal constituent of the HIV capsid protein (CA) exhibits behavior similar to a karyopherin, forming multivalent, weak, and specific associations with the FG repeat domains in the NPC's permeability barrier. This direct interaction is suggested to be sufficient to overcome the permeability barrier, allowing the viral capsid to traverse the NPC and gain access to the nucleus in non-dividing human cells.

Methodologically, they first demonstrate that monomeric and oligomeric CA directly interact with various forms of FG domains *in vitro*, dependent on the integrity of the CA FG-binding pocket.

Probably, this comment was meant for the parallel study by Dickson and colleagues. We do not show any data on monomeric CA interacting with FG domains, as it is well established that the complete FG binding site forms through CA hexamerization (see Price *et al.*, 2014). Consistent with that, we show that a monomeric CA-GFP fusion shows no discernable NPC binding while fully assembled capsid spheres target NPCs very efficiently (see Fig. 1b).

Next, they use reconstituted FG domain condensates *in vitro* to show that capsid-like structures of CA effectively interact with FG elements of the condensates and can even infiltrate these condensates to distances corresponding to the length of the NPC central channel. The viral-like particle's interaction and merging with the condensates does not destabilize the CA oligomers, nor necessitate the presence of nuclear transport factors or RanGTP. The authors posit that the karyopherin-like arrangement of CA within the capsid lattice endows the capsid with self-translocating characteristics, accounting for the virus's ability to infiltrate the nucleus of non-dividing cells.

This is an incomplete and inexact account of our findings. We do not posit, rather, we demonstrate NTR-like properties of the capsid. It enters the FG phase with partition coefficients that are *on par* with traditional NTRs. We further demonstrate that the capsid can stably enclose a cargo and carry it into the phase, resulting in a partition coefficient that is $\sim 10\,000$ times higher than that of the free cargo. By these criteria, the assembled capsid is a nuclear transport receptor. Moreover, we show that the capsid targets itself not only into a pure FG phase but also to human NPCs, reaching essentially complete occupancy (see new Fig. 1c).

However, in comparison with the current state of the field, this study has significant shortcomings. Much is already known about how CA interacts with the NPC and how the HIV capsid traverses the NPC, and this work does not significantly advance our mechanistic understanding of this process. It's widely recognized that CA directly interacts with FG domains, as evidenced by multiple research groups and reviewed by researchers such as Kane et al., 2018 and Guedan, 2021. Additionally, structures illustrating the exact manner in which CA binds FG repeats have been published (e.g., Price et al., 2014; Stacey et al., 2023), detailing how a specific pocket attaches to the FG repeat. As such, it's predictable that altering this pocket would impede this interaction. Hence, the molecular mechanism of capsid interaction with FG repeats to enter the NPC is well-known.

Our introduction gave an accurate account of the current state of the field. We cite that various FG-Nups are recognized by CA. However, there is debate as to whether specific, individual FGs are important for HIV-1 import, or, as we and the Dickson et al. study now argue, that it is a general affinity for FGs across the entire NPC that facilitates capsid transport. For instance,

the 1407-1423 region of Nup153, including a single FG peptide, and its specific function has been a focus of the field (Matreyek *et al.*, 2013; Buffone *et al.*, 2018; Shen *et al.*, 2023). Prominent current models propose a direct hand-over from the cytoplasmic side (Nup358) of the NPC to Nup153 on the nucleoplasmic side (Zila *et al.*, 2021; Shen *et al.*, 2023), without invoking the general FG-interaction with many Nups that we propose. In addition, none of the previous studies addressed the highly dynamic, NTR-like interaction of CA-hexamers and capsid-like particles that we and the Dickson *et al.* study now demonstrate. It was also not known that these interactions are productive in the sense that they confer an efficient partitioning of the capsid in an otherwise extremely restrictive FG phase barrier (that excludes inert molecules with 1000 times smaller volumes). The latter is not a trivial point. Indeed, the ΔG of all FG interactions must compensate for the energetic penalty for counteracting surface tension of the phase and local cohesion between FG repeats and for creating a cavity inside the phase that is large enough for accommodating the giant capsid.

Furthermore, the FG interactions of the mobile species must be balanced and properly distributed over its surface. We demonstrated before that such balance is not trivial either (see Ribbeck and Görlich, 2002; Schmidt and Görlich, 2015; Ng *et al.*, 2023), with a striking experiment showing that anti-FG repeat antibodies bind only to the surface of an FG phase but are unable to enter in (see supplemental figure 2 in Frey and Görlich, 2009).

CA is known to interact with various other cellular proteins that facilitate NPC interaction. It's established that the FG binding pocket interacts with several non-Nup proteins like CPSF6 and Sec24C. Furthermore, CA also interacts with the nuclear transport factor TNPO3 and the nucleoporin Nup358, not through its FG repeats but its cyclophilin-like domain. Despite acknowledging these known interactions, the authors overlook these additional host co-factors demonstrated to be involved in capsid entry to the NPC in their mechanistic models. Given the known dynamic nature of these interactions, asserting that adding them to the entire capsid would create an oversized shell unable to pass through the NPC is an unsupported assumption. There could be various arrangements and substoichiometric associations that enable passage, considering these proteins are known to aid capsid entry into the NPC. As per Guedan, 2021, "the viral core should be considered a dynamic structure that binds numerous cellular proteins on its path through the NPC," a contrasting view from that presented in this manuscript.

These points do not conflict with our main story. We demonstrate sufficiency. The capsid alone is sufficient to penetrate into an otherwise strict permeability barrier. We cannot see that this discovery would be questioned in any way by the details listed above.

For the specific points: TNPO3 (transportin SR) is a typical importin that releases cargo upon contact with RanGTP (Kataoka *et al.*, 1999). If TNPO3 were targeting the capsid into the central NPC channel, then this would have been abolished by the dominant-negative RanGTP Q69L Δ C mutant. We observed, however, RanGTP-resistant NPC targeting of the capsid (see Figure 1), excluding targeting by any importin. In fact, the requirement of TNPO3 for HIV infection is well explained by mediating nuclear import of CPSF6, the mislocalization of which is known to preclude nuclear entry of HIV-1 (De Iaco *et al.*, 2013; Maertens *et al.*, 2014; Jang *et al.*, 2019). Our work cannot speak to release mechanisms from the NPC, only targeting to the NPC and partitioning as far as the basket.

The cyclophilin homology domain of Nup358 (CypHD) is located in the cytoplasmic filaments of NPCs. We do not question its interaction with the capsid, however, if this were the complete story then this would only explain capsid-targeting to the cytoplasmic filaments. Indeed, in our BLI assay, we show interactions between CA hexamers and FG-regions of Nup358, which are distinct from the CypHD. It is also known in the field that removing CypHD from Nup358 does

not abolish HIV-1 infection (Meehan *et al.*, 2014). Additionally, Zila and colleagues show the HIV-1 capsid fully inserted into the central channel via cryo-electron tomography. We suggest that the capsid is held there by direct interactions with FG repeats that are present at concentrations of > 100mg/ml. Our demonstration that the capsid is literally sucked into any transport-competent FG phase (Figures 3-4) has been the most direct test of this assumption.

Key questions remaining in the field about this process are also not really addressed. These include questions on the sequence of events, possible NPC dilation or re-structuring, the roles of nuclear basket, Nup88 complex and cytoplasmic filaments, the coupling of viral entry to the NPC and uncoating, and the uncertainty of whether intact capsids completely traverse the NPC or disassemble during the process. Zila *et al.*, 2021 illustrated that the capsid enters the NPC narrow end first, suggesting NPC dilation to accommodate it, thereby pointing towards an orderly entry mechanism; their EM analysis also noted major remodeling of capsids upon NPC entry, corroborated by other groups' research. None of these pivotal steps are addressed in this work, indicating that the model presented here is inadequate.

A comprehensive model of HIV-1 infection also includes receptor-binding, cell entry, reverse transcription, genome-integration, immune evasion, etc. Addressing all of these areas was not the purpose of this study. We focused on a single, central point, namely on how the capsid overcomes the FG phase-based permeability barrier of NPCs. This is a key question in the field which has not been fully addressed. Indeed, Dickson *et al.* have also investigated this question, independently, resulting in similar conclusions, however, it is clear this is an open question in the field. Our work also addresses questions important to the NPC field outside of HIV: the ~60 nm diameter of the human NPC is a relatively new finding (Zila *et al.*, 2021; Schuller *et al.*, 2021), and the ability of the NPC to transport cargoes which approach its size limit is an area of interest. From our perspective, the above points do not render our focused analysis inadequate nor do they negate our central finding. What the reviewer notes are simply separate questions, which, while interesting, we did not intend to interrogate with our study.

Moreover, the conclusions that are presented are not fully substantiated by the data presented (below). Hence, I believe they this manuscript is not suitable for publication in Nature. My other main general concerns are as follows:

1. The authors assume that FG macroscopic condensates replicate the properties and behavior of the nanoscale NPC permeability barrier, a premise that is not universally accepted. In fact, this assumption and in vitro system were recently rebutted in a publication in Nature (Yu *et al.*, 2023) and a BioRxiv publication (Kozai *et al.*, 2023), which collectively demonstrate that the behavior of FG domains in condensates does not accurately reflect the state and properties of native, NPC anchored FG Nups. Thus, many of the conclusions in this paper – based on this other work – could be incorrect. The authors do not address this serious issue.

With all due respect to the reviewer, everybody is entitled to their own opinion, but not to their own facts. Yu *et al.*, 2023 (a study from Ed Lemke's lab) agree with us that Nup98 condensates indeed show very similar transport selectivity as NPCs. Kozai *et al.*, 2023 (a study from Roderick Lim's lab), a non-peer reviewed pre-publication, grossly misrepresents the FG phase literature. But apart from that, it concedes in its discussion that FG condensates do indeed show NPC-like properties. At the end of the day, it is reasonable to say that the assay shows NTR-ness. Non-valid cargoes are not able to partition into these condensates. NTRs do. HIV-1 capsid assemblies likewise partition into the condensates, a behavior only observed in NTRs. Thus, by the standards of the assay, HIV-1 capsids are indeed NTR-like.

2. Based on the data presented, I believe a more straightforward and adequately substantiated interpretation would be that the CA capsid lattice might facilitate the docking of the HIV capsid to the NPCs. However, the manuscript does not provide convincing proof that the HIV capsids or the smaller 40 nm capsid-like spheres fully traverse the NPC *in vivo*. The authors mention this idea, but they don't consider it a primary conclusion, instead favoring a notion of Kap-like trafficking through the NPC, a possibility that remains unverified. The integration into FG condensates isn't sufficient proof, and it would be beneficial to demonstrate this in an actual NPC.

We do indeed demonstrate that the capsid inserts with an amazing efficiency into actual NPCs, without the help of soluble factors. In fact, the occupancy of NPCs with capsid spheres is virtually 100% (see Figure 1, and in particular the new panel c). Within one hour of incubation, the capsid diffused μm -distances into the FG phase and would thus cross the distance of the central NPC ($\sim 100\text{ nm}$) within less than 1/100 of this time. This is well in range of what to expect. Once inserted there, the capsid will sit in an energy well and it is indeed unclear how it is released from there.

A likely possibility, shared by Reviewer 3, is that nuclear factors (e.g., CPSF6) compete out the FG-Nup interaction and thus release the capsid into the nucleoplasm. In fact, we find that 40 nm capsid spheres microinjected into the cytoplasm of mouse oocytes readily cross NPCs, enter the nucleoplasm and accumulate at intra-nuclear foci. Please see new Extended Data Figure 5.

3. Karyopherins form multivalent, weak, and transient interactions with the FGs in the NPC. It remains unclear from the evidence provided that this is how CA multimers in the HIV capsid operate. Thus, without further data and controls, the comparison of CA to a karyopherin, though intriguing, may not accurately depict the CA lattice's actual behavior and could potentially mislead the general reader.

It is true, nuclear transport receptors form multivalent, weak, and transient interactions with FGs. Biolayer interferometry is a sensitive assay ideal for capturing such interactions, and our study shows that FG-binding to HIV-1 capsid hexamer (Fig. 2a) is highly reminiscent of the binding mode of an NTR to these same FGs (Fig. 2b). Thus, we feel it is appropriate to draw the comparison between NTR-FG interactions and capsid-FG interactions. Additionally, the N57A mutation in the FG-binding pocket of capsid hexamers reduces binding, further demonstrating that, like NTR-FG interactions, the engagement with FGs is direct. We welcome solid scientific suggestions from the reviewer for experiments or controls that will bolster our claims, but it is not fair to say that we are misleading general readers. We note that this phrasing was used verbatim in the review of Dickson *et al.* Our study makes a direct, evidence-based comparison between the binding modes of NTR-FG interactions and capsid-FG interactions, so it is unclear why this is an appropriate criticism of our manuscript, though, again, if the reviewer has not found Fig. 2 convincing, we welcome suggestions for additional analyses.

4. The manuscript does not clearly state the actual duration of the experiments performed with these condensates. These are known, even by the authors' admission, to be susceptible to "aging", gelation, and formation of potentially aberrant beta-structures, a scenario unlikely to occur in a living NPC permeability barrier. How would the "aging" of the condensates influence the different experiments presented in these manuscripts?

This statement inaccurately reflects what we wrote in our manuscript. It appears to be a slightly reworded comment originally intended for the Dickson *et al.* study.

We show one hour's time points - clearly described in the legends.

‘Aging’: The effect has been described for condensates that eventually form amyloid structures - heavily studied for (pathogenic forms of) Fus. Amyloids can clearly be detected by Thioflavin T staining and by their characteristic NMR signatures (originating from stable cross β -sheets). This is not an issue for the GLFG phase used here, as documented in our previous publications (see Figure 2 in Ng *et al.*, 2021; Figure 6 in (Schmidt and Görlich, 2015), and the NMR analysis of Najbauer *et al.*, 2022).

An amyloid propensity of FG repeat domains can be seen, for example, in the extremely NQ-rich Nup100 and Nup116 FG domains from *S. cerevisiae*, which are indeed related to the yeast prion Sup35 (see for example Michelitsch and Weissman, 2000; Ader *et al.*, 2010; Schmidt and Görlich, 2015), and in engineered cohesive FG domains depleted of prolines (see Figure 2 in Ng *et al.*, 2021).

The ‘perfect’ GLFG repeats used here have a normal proline content and do not show amyloids/cross- β structures by any analytical method – not even in months old NMR samples (Najbauer *et al.*, 2022). Otherwise, we agree with the reviewer in this point that aberrant cross- β structures are highly unlikely to contribute to the NPC’s permeability barrier.

5. It's concerning that the presence of karyopherins seems to significantly outcompete the CA protein or hexamer in its interaction with FG condensates. This situation would be more akin to what an HIV capsid would encounter in a living NPC: hundreds of copies of different types of karyopherins and other nuclear transport factors (carrying RNPs, Ran, etc), all interacting and passing through the NPC along with their cargoes and other molecules, creating a heterogeneous, complex, and dynamic nano-environment. The capsid would never encounter pure and homogeneous FG condensates. How do the authors reconcile these seemingly conflicting observations? The authors should show the effect that the presence of excess karyopherins have in their FG condensate CA interaction assays, more accurately reflecting *in vivo* conditions.

This statement inaccurately describes our findings and is apparently just slightly edited from a comment intended for the Dickson *et al.* study. We do not show karyopherins/NTRs outcompeting CA assemblies in our studies.

But, to address the point: The EM tomograms of Zila *et al.* 2021, showing capsids inside NPCs, make it very plausible that the capsid displaces any (larger) cargo from the central channel. The capsid is likely to be a very competitive FG phase client, as when it is present, little space remains. This would explain why its NPC targeting is so effective (Figure 1 and below). Indeed, we observed that capsid entry into the GLFG phase resists competition by ~physiological concentrations of traditional NTRs (see Figure below). This can be explained by the extraordinary high valency of FG binding sites in the capsid.

Information for Editors/ reviewers. Partitioning GFP-labelled capsid spheres into the GLFG phase was probed as in Figure 3. Where indicated, 2 μ M of the respective NTRs had been added. Note that these neither enhanced nor inhibited phase entry of the capsid. Also note that this concentration is already slightly higher than the respective cellular concentrations of these NTRs.

6. The fact that the capsid spheres localize at the nuclear rim of digitonin-permeabilized HeLa cells suggests that they might indeed be docking to NPCs, however, electron microscopy could be used to verify if that's the case. Such an experiment would nicely demonstrate that the particles are not just attached to the NE and could potentially illustrate how those particles interact with the NPC. Are they attached to the periphery? Blocked in transit through the channel? Retained at the nuclear side? Do they show a continuum of all these behaviors? It would strongly support the authors' claim if they could show and analyze the attachment and localization of those particles in native, active NPCs, also nicely complementing their in vitro assays with an in-situ analysis.

We could do EM - though this will take quite some time and the outcome will likely be the same as the tomograms of Zila *et al.*. We have, however, recorded higher resolution fluorescence images that are consistent with the Zila *et al.* data and with the capsid spheres having entered the central NPC channel (new Figure 1c). Furthermore, our new data mouse oocyte data clearly indicate that the same capsid preparation can complete the NPC passage and reach the nuclear interior (see new Extended Data Fig. 5).

7. The localization of the capsomeres and the CA-spheres in the FG condensates is quite heterogeneous. The capsomeres seem to accumulate at the rim of the big condensates, but in the smaller ones, they do seem to penetrate further inside (see Ext.Data Fig. 3d), showing a much thicker rim stain, is this caused by an optical effect or is it a real property dependent on the size of the condensate?

Not all the FG particles are in the same plane. A surface signal appears in an equatorial scan like a thin rim. If the focal plane is focused on the surface, one sees a smaller filled circle. The particles in question are in between.

In the case of the CA-spheres, there are particles in which most of the signal seems to concentrate at the rim of the condensate, and in other penetrates non-homogeneously in the condensate, even forming clear bright concentrated regions within the condensates. Could this be caused by the formation of amyloids in the condensates as they age? The condensates are presumably not homogeneous, as shown recently in the Lim lab BioRxiv paper - I would like to ask the authors to comment and explain these effects.

Rim-staining: this might again be a comment for the parallel paper. The rim-staining is most obvious for the hexameric capsomers (fused to the super-inert GFP sinGFP4a), while capsid spheres and CLPs enter the here analyzed FG phases completely (see e.g., Figure 3b for a comparison). Acquisition of full NTR-properties depends on complete capsid assembly, as described and discussed in our manuscript.

As explained above, the perfect GLFG phase shows no aging effects or amyloid propensity whatsoever. The non-homogeneous distribution of the capsid spheres is due to its slow diffusion, starting at the surface. We also see the phenomenon that smaller particles give a brighter signal, which is plausible because they have a larger relative surface to absorb the capsid spheres and because it takes less time to diffuse to their centers. Occasionally, we see fusions between small and larger FG particles. Then it takes some time to equilibrate the intra-phase concentrations of the capsid.

The Lim lab focused on the FG domain of yeast Nup100, which is extremely NQ-rich, and in that related to the prion Sup35. By the criterion of its bright staining with Thioflavin T, it has the highest amyloid propensity of all analysed Nup98 homologs so far (Schmidt & Görlich, 2015). Why Roderick Lim and this reviewer generalize from Nup100 to other FG domains is not quite clear to us, in particular as we were transparent about the used experimental system and wrote: *“We also chose this model because it avoids complications like O-glycosylation or amyloid formation, and because it is very well characterized and known to faithfully recapitulate importin- and exportin-mediated cargo transport, response to the RanGTPase system, as well as NTF2-mediated retrieval of RanGDP to nuclei.”*

Referee #3 (Remarks to the Author):

Fu and colleagues study FG peptide binding to CA hexamers by BLI and from here study the ability of CLPs (spheres and cone-like) to enter into artificial FG hydrogels. The data is consistent with the interpretation that the HIV-1 capsid is likely to serve as its own NTR.

One problem with this paper is apparent mis-citing between text and figure panels. This surely complicates data interpretation for readers. This moreover gives the impression of a rush job. One is struck by a somewhat similar though more comprehensive study available on bioarchives, which possibly impacted compilation speed here. Nevertheless, this paper incorporates the unique control to incorporate a soluble fluid marker into CLPs and show that this marker stays incorporated following hydrogel intrusion.

The paper has no virology data and the authors seem to have limited virology experience. This is evident by some blatant mis-statements. The authors also cite only some of the relevant HIV literature.

Apologies and thank you for your thorough proof-reading!

1) The authors target five Nup proteins for study Ext Data Fig 2b. Have any of these proteins been shown previously to bind CA? Have knockdown cells been challenged with HIV? What are the results of these experiments? Are these data consistent with roles for these Nups in HIV nuclear import?

Human NPCs contain ~10 different FG Nups. Given their different anchoring positions, they probably form different FG layers along the path through NPCs. It seems very likely to us that any mobile species crossing NPC will 'see' all or most of them during transit.

Interpreting the effects of an FG Nup knockdown on infection is not really straightforward. At least the longer FG domains will have a barrier function and thus suppress passage of the capsid through NPCs. So, the elimination of a barrier element could favor capsid entry. However, one can also imagine scenarios where such elimination is detrimental for infection. Furthermore, several FG Nups are part of obligate complexes (Nup62, 54, 58, 214), part of the NPC scaffold (Nup98, Nup358), essential for FG-unrelated reasons (e.g. Nup358, Pom121), or required for anchoring the nuclear basket (Nup153).

Binding of various FG Nups has been shown before. For example, Nup98 and Nup153 have been shown to bind CA-NC, as we cite in Di Nunzio *et al.*, 2013. Other studies used a co-pelleting assay with CA-nanotubes. Some of these studies were performed with whole cell lysates and detected by immunoblotting, thus not confirming direct interaction (Kane *et al.*, 2018). None of the previous studies focused on dynamic, direct interaction the way we and the Dickson study now do. Our fragment-based screen also shows that, even for already-studied FG Nups, the interaction is unlikely to be linked to one specific FG motif; instead, binding to FGs is more general.

We adjusted text and references to reflect prior data more thoroughly.

2) Page 3 line 2. Retrovirus genetic material is RNA. It is the DNA form made by reverse transcription that is integrated.

True, this was misleading and is corrected now to read: 'To establish infection, retroviruses must integrate a DNA copy of their reverse transcribed RNA genomes into host chromosomes.'

3) 4 lines from bottom. HIV is not endocytosed. Its membrane fuses with the cellular plasma membrane.

Corrected as suggested to read now 'Early steps of HIV-1 infection, namely surface receptor binding and membrane fusion, ensure the delivery of the viral capsid to the cytoplasm of the target cell ...'

4) Please cite the IBB control experiment Fig 1c in main text page 4.

Done.

5) Page 5 line 1, “and CPSF6 peptide controls (Extended Data Fig 2c)”

Done.

6) Line 2, cite Fig 2c after “CA-N57A mutation”

Done.

7) Line 3, cite (Extended Data Fig 2c) at end of sentence.

Done.

8) Line 4, Fig 2c should apparently be Fig 2b.

Corrected.

9) 2nd full paragraph line 3, for clarity, please indicate (Fig. 3d).

We cite the entire Fig.3 as this refers to panels a, b, c, and d.

10) Ending sentence “But still, ...”. Where is this data shown?

in Fig.3a, but this applies also to later panels. Considering the flow of text, we cite Fig. 3a.

11) Page 6 line 4, please add citations after “the capsid spheres.”

Reference 10 refers to an earlier publication, where we characterized the FG domains used in experiment of Fig. 3c. That is why this citation is placed after ‘motifs’.

12) Next paragraph line 4, please show separation of free mCherry by gel filtration.

We added this as new Extended Data Figure 6.

13) Line 8, Fig 3b should apparently be Fig 3d?

The exclusion of sinGFP4a is also shown in Fig. 3d. This sentences, however, referred indeed to 3b (note that the numberings of figures have changed now).

14) 6 lines from bottom, Fig 3c and Ext Data Fig 3b should apparently be Fig 3d and Ext Data Fig 3c. Also, shouldn't at least 500 be “approximately 26,000”?

To calculate the partition coefficient, one needs to divide 26 000 by 50. It is the ratio of inside: outside signals. To improve clarity, we now added the partition coefficients directly in the figures.

15) Page 7 middle paragraph line 2, please also cite ref 24, which first showed this for Nup153.

Done as suggested.

16) Please avoid “It is not exactly known if the capsid ever completes NPC passage”. Ref 15, 16 images convincingly show incursion into the nuclear lumen before apparent HIV uncoating for integration. Moreover, viruses unable to bind CPSF6 (through knockdown or CA mutation) arrest at the NPC and integrate into novel genomic regions at the nuclear periphery. Thus, while CPSF6 deficient viruses seem perhaps unlikely to complete NPC passage, there is solid evidence to suggest otherwise under basal infection conditions.

We have re-written this along these lines. In fact, the mouse oocyte data now included as Extended Data Figure 5 are consistent with capsids completing NPC passage and reaching the nucleoplasm.

17) Page 9 Fig 1b legend seems to omit capsomere description. Line 6, “life” should be “live”.

We used here a monomeric CA-EGFP fusion. The lettering is now corrected, and the legend completed. “life” was changed to “live”.

18) Final word page 9, “equally” is gross over-statement. For Nup58, there is only 2-fold reduction, while affects onCPSF6 and Nup98 seem 10-fold or greater. Nup62 is somewhere in between.

True. And perhaps functionally relevant in the sense that the capsid being able to bind different FG repeats in different ways. The sentence was re-written along these lines.

19) Page 11 line 1, as meant? Doesn't seem that CA-mEGFP is in this figure.

True. Corrected.

20) Please avoid CypA label at Nup358 C-terminus, Ext Data Fig 2b. There are nearly 20 human cyclophilins, and CypA is a standalone protein. Consider CypH for cyclophilin homology domain.

Changed as suggested.

Arbitrating reviewer (4).

Regarding novelty and advance, the arbitrating referee tended to agree with our more critical ref. This referee also felt the advance was somewhat incremental, particularly in light of a recently published paper in Nature Communications (<https://www.nature.com/articles/s41467-023-39146-5>). The reviewer was also puzzled by a disagreement between your work and the work presented in that paper, where knocking down Nup98 didn't have much effect, while knocking down Nup35 had a more pronounced effect. That paper must be cited in the revision and the discrepancy addressed.

The mentioned Nature Communication paper is less relevant to our study than it might appear from its misleading title that is actually not well supported by the data. In fact, this paper does not provide any evidence for the HIV-1 capsid traversing the NPC or an FG phase in a nuclear transport receptor-like manner. There is not a single mentioning of Nup98 in the main text. Nup98 was a target in an RNAi screen, but there was neither an effect on infection nor any validation of a successful knockdown. Instead, this paper focused on Nup35.

Here, it is unclear why the authors of that paper reported an interaction of the capsid and Nup35 FG repeats. Nup35 is a structural component of the inner NPC ring and does not comprise an FG repeat domain. While Nup35 does contain three FG dipeptides, none of them is in a context of an intrinsically disordered, low-complexity sequence. The first FG dipeptide (Nup35¹⁷⁹⁻¹⁸⁰) is buried in the RRM dimer interface. The second (Nup35²³²⁻²³³) is also part of the globular RRM fold. The C-terminal FG-motif (Nup35³²⁴⁻³²⁵) is part of a membrane-binding motif that contributes to anchoring NPCs within the nuclear envelope. None of these FG dipeptides is likely to interact with any translocating species.

Comparison between Nup35 and Nup98: Nup35 and Nup98 are both essential for NPC biogenesis (Nup98 not only contributes to the permeability barrier but also links structural Nups, such as Nup155, Nup188, and Nup96 to each other). Thus, any complete knockdown would imply the absence of NPCs. It is unclear how such an experiment could reveal how the HIV-1 capsid crosses NPCs. As an absence of NPCs would be lethal, only partial knockdowns can be analysed in infection experiments.

Assuming, an incomplete knockdown would result in NPCs with less Nup98 molecules than usual, this would result in a more permissive (more leaky) permeability barrier. Would one predict it becomes more difficult for the capsid to cross such a weakened barrier and that a Nup98 knockdown would specifically impede infection? Certainly not. In summary, we cannot see a discrepancy between their and our datasets and we do not think that it would be helpful to readers if we included the above (rather distracting) discussion into our manuscript. Likewise, we cannot just include the citation without pointing readers to the problems and thereby propagate errors of fact. This is something to discuss in a review.

Regarding physiological relevance, the arbitrating expert was somewhat equivocal on the question of whether in vitro FG condensates are sufficient to recapitulate the in vivo NPC, but the expert did acknowledge that they can be useful transport models, when used to complement cell-based studies. We would therefore ask you to include the additional data in mouse oocytes and HeLa cells to which you referred in your rebuttal, particularly as this new expert also felt that your figure 1 was not convincing, as it lacks any control where the capsid does not stick to the nuclear envelope.

The finding that NPC targeting of the capsid does not require the addition of trans-acting factors is a key conclusion of our study and not a shortcoming.

Furthermore, we provide controls for specificity: (i) we observed a very convincing colocalisation with NPCs and not just a nuclear envelope binding, (ii) the prominent NPC binding is dependent on capsid assembly (compare the binding of capsid monomers with that of 40 nm capsid spheres in Figure 1b), (iii) the permeabilized cells contain not only NPCs but also mitochondria, bulk ER, other vesicles and membrane systems, the cytoskeleton, etc. NPCs account there for less than 0.1% of the cell's volume, and yet they accumulated >50% of the capsid signal (with most of the remainder probably representing annulate lamellae). This is a highly specific targeting event, with a molecular mechanism we decipher in the FG phase experiments.

As requested, we now include the higher resolution colocalization of the capsid with HeLa NPCs (as new Figure 1c) as well as the mouse oocyte injection experiment (as Extended Figure 5) in the new version of manuscript.

Additional points that the expert raised are as follows:

- The 40nm spherical capsid is very different from the HIV cone shaped capsid, which is 70nm at its widest point. How does the widest part make it through if indeed it does? Some discussion of this seems necessary, and the limitations of the 40nm sphere should be clear to the reader.

We directly compared the 40 nm capsid spheres with the much larger (60 x 150 nm) capsid-like particles (CLPs) and found that the two show the same very efficient partitioning into the very dense GLFG phase. This suggests that both capsid species have the same surface properties and the same NTR-like behavior. Given the rather large difference in size, one might expect that the larger species would have more problems in entering the phase. However, this size effect is fully compensated by the proportionally larger number of FG contacts of the CLPs. This is discussed in the paper (see also below).

Our main conclusion is that the capsid has NTR-like properties, and to this end we further show that the 40 nm capsid spheres target authentic NPCs with high specificity and without the help of trans-acting factors.

Of course, the size difference between spheres and CLPs (or the authentic HIV-1 capsid) does matter in respect to passage through the NPC scaffold (given its geometric constraints). However, Hans-Georg Kräuslich's and Martin Beck's groups have reported before that the HIV-1 capsid can fully insert into NPCs. We therefore take this as an established fact and cite it as published data.

- The reviewer felt that this sentence was overinflated: "Considering that the spheres are very large in mass (6 MDa) and diameter (40 nm), this efficient entry might appear surprising. However, this can be explained by (i) the cooperation of...". The expert felt that this wasn't surprising as entry of large capsids has been seen before.

We did not write "*it is surprising*", we wrote "*Considering that ... it might appear surprising*", which is quite different. Furthermore, there is no previous report of a viral capsid entering a dense FG phase without the help of a trans-acting nuclear transport receptor. We therefore feel that our phrasing is perfectly appropriate. Furthermore, the paragraph is necessary as it connects the concept of size exclusion from an FG phase (whereby the ΔG for exclusion scales with size

or surface area – to be precise) with the concept of multivalency of FG interactions (whereby the ΔG of attraction scales with the number of binding sites).

To avoid misunderstandings, we slightly expanded the paragraph to read:

NPCs have long been known to act like sieves, with exclusion scaling with the size of the mobile species (Bonner, 1975). Considering that the spheres are very large in mass (6 MDa) and diameter (40 nm), their efficient FG phase entry might appear surprising. However, this can be explained by (i) the cooperation of 240 FG binding sites on the capsid sphere's surface and (ii) by the burial of FG-repellant elements (Frey et al., 2018), such as the relatively charged CA 'interior' and the C-terminally fused GFP.

Cited references

- Ader C, Frey S, Maas W, Schmidt HB, Görlich D, Baldus M (2010) Amyloid-like interactions within nucleoporin FG hydrogels. *Proc Natl Acad Sci USA*, **107**: 6281–6285. doi: 10.1073/pnas.0910163107
- Bonner WM (1975) Protein migration into nuclei. I. Frog oocyte nuclei in vivo accumulate microinjected histones, allow entry to small proteins, and exclude large proteins. *J Cell Biol*, **64**: 421–430. doi: 10.1083/jcb.64.2.421
- Buffone C, Martinez-Lopez A, Fricke T, Opp S, Severgnini M, Cifola I, Petiti L, Frabetti S, Skorupka K, Zadrozny KK, Ganser-Pornillos BK, Pornillos O, Di Nunzio F, Diaz-Griffero F (2018) Nup153 Unlocks the Nuclear Pore Complex for HIV-1 Nuclear Translocation in Nondividing Cells. *J Virol*, **92**: e00648–18. doi: 10.1128/JVI.00648-18
- De Iaco A, Santoni F, Vannier A, Guipponi M, Antonarakis S, Luban J (2013) TNPO3 protects HIV-1 replication from CPSF6-mediated capsid stabilization in the host cell cytoplasm. *Retrovirology*, **10**: 20. doi: 10.1186/1742-4690-10-20
- Di Nunzio F, Fricke T, Miccio A, Valle-Casuso JC, Perez P, Souque P, Rizzi E, Severgnini M, Mavilio F, Charneau P, Diaz-Griffero F (2013) Nup153 and Nup98 bind the HIV-1 core and contribute to the early steps of HIV-1 replication. *Virology*, **440**: 8–18. doi: 10.1016/j.virol.2013.02.008
- Frey S, Görlich D (2009) FG/FxFG as well as GLFG repeats form a selective permeability barrier with self-healing properties. *EMBO J*, **28**: 2554–2567. doi: 10.1038/emboj.2009.199
- Frey S, Rees R, Schünemann J, Ng SC, Fünfgeld K, Huyton T, Görlich D (2018) Surface properties determining passage rates of proteins through nuclear pores. *Cell*, **174**: 202–217.e9. doi: 10.1016/j.cell.2018.05.045
- Jang S, Cook NJ, Pye VE, Bedwell GJ, Dudek AM, Singh PK, Cherepanov P, Engelman AN (2019) Differential role for phosphorylation in alternative polyadenylation function versus nuclear import of SR-like protein CPSF6. *Nucleic Acids Res*, **47**: 4663–4683. doi: 10.1093/nar/gkz206
- Kane M, Rebersburg SV, Takata MA, Zang TM, Yamashita M, Kvaratskhelia M, Bieniasz PD (2018) Nuclear pore heterogeneity influences HIV-1 infection and the antiviral activity of MX2. *Elife*, **7**: e35738. doi: 10.7554/eLife.35738
- Kataoka N, Bachorik JL, Dreyfuss G (1999) Transportin-SR, a Nuclear Import Receptor for SR Proteins. *The Journal of Cell Biology*, **145**: 1145–1152. doi: 10.1083/jcb.145.6.1145
- Kozai T, Fernandez-Martinez J, van Eeuwen T, Gallardo P, Kapinos LE, Mazur A, Zhang W, Tempkin J, Panatala R, Delgado-Izquierdo M, Raveh B, Sali A, Chait BT, Veenhoff LM,

- Rout MP, Lim RYH (2023) Dynamic molecular mechanism of the nuclear pore complex permeability barrier. *bioRxiv*, 2023.03.31.535055. doi: 10.1101/2023.03.31.535055
- Maertens GN, Cook NJ, Wang W, Hare S, Gupta SS, Öztop I, Lee K, Pye VE, Cosnefroy O, Snijders AP, KewalRamani VN, Fassati A, Engelman A, Cherepanov P (2014) Structural basis for nuclear import of splicing factors by human Transportin 3. *Proc Natl Acad Sci U S A*, **111**: 2728–2733. doi: 10.1073/pnas.1320755111
- Matreyek KA, Yücel SS, Li X, Engelman A (2013) Nucleoporin NUP153 phenylalanine-glycine motifs engage a common binding pocket within the HIV-1 capsid protein to mediate lentiviral infectivity. *PLoS Pathog*, **9**: e1003693. doi: 10.1371/journal.ppat.1003693
- Meehan AM, Saenz DT, Guevera R, Morrison JH, Peretz M, Fadel HJ, Hamada M, van Deursen J, Poeschla EM (2014) A cyclophilin homology domain-independent role for Nup358 in HIV-1 infection. *PLoS Pathog*, **10**: e1003969. doi: 10.1371/journal.ppat.1003969
- Michelitsch MD, Weissman JS (2000) A census of glutamine/asparagine-rich regions: implications for their conserved function and the prediction of novel prions. *Proc Natl Acad Sci U S A*, **97**: 11910–11915. doi: 10.1073/pnas.97.22.11910
- Najbauer EE, Ng SC, Griesinger C, Görlich D, Andreas LB (2022) Atomic resolution dynamics of cohesive interactions in phase-separated Nup98 FG domains. *Nat Commun*, **13**: 1494. doi: 10.1038/s41467-022-28821-8
- Ng SC, Biswas A, Huyton T, Schünemann J, Reber S, Görlich D (2023) Barrier properties of Nup98 FG phases ruled by FG motif identity and inter-FG spacer length. *Nat Commun*, **14**: 747. doi: 10.1038/s41467-023-36331-4
- Ng SC, Güttler T, Görlich D (2021) Recapitulation of selective nuclear import and export with a perfectly repeated 12mer GLFG peptide. *Nat Commun*, **12**: 4047. doi: 10.1038/s41467-021-24292-5
- Price AJ, Jacques DA, McEwan WA, Fletcher AJ, Essig S, Chin JW, Halambage UD, Aiken C, James LC (2014) Host cofactors and pharmacologic ligands share an essential interface in HIV-1 capsid that is lost upon disassembly. *PLoS Pathog*, **10**: e1004459. doi: 10.1371/journal.ppat.1004459
- Ribbeck K, Görlich D (2002) The permeability barrier of nuclear pore complexes appears to operate via hydrophobic exclusion. *EMBO J*, **21**: 2664–2671. doi: 10.1093/emboj/21.11.2664
- Schmidt HB, Görlich D (2015) Nup98 FG domains from diverse species spontaneously phase-separate into particles with nuclear pore-like permselectivity. *Elife*, **4**: e04251. doi: 10.7554/eLife.04251
- Shen Q, Feng Q, Wu C, Xiong Q, Tian T, Yuan S, Shi J, Bedwell GJ, Yang R, Aiken C, Engelman AN, Lusk CP, Lin C, Xiong Y (2023) Modeling HIV-1 nuclear entry with nucleoporin-gated DNA-origami channels. *Nat Struct Mol Biol*, **30**: 425–435. doi: 10.1038/s41594-023-00925-9
- Yu M, Heidari M, Mikhaleva S, Tan PS, Mingu S, Ruan H, Reinkemeier CD, Obarska-Kosinska A, Siggel M, Beck M, Hummer G, Lemke EA (2023) Visualizing the disordered nuclear transport machinery in situ. *Nature*, **617**: 162–169. doi: 10.1038/s41586-023-05990-0
- Zila V, Margiotta E, Turoňová B, Müller TG, Zimmerli CE, Mattei S, Allegretti M, Börner K, Rada J, Müller B, Lusic M, Kräusslich HG, Beck M (2021) Cone-shaped HIV-1 capsids

are transported through intact nuclear pores. *Cell*, **184**: 1032–1046.e18. doi:
10.1016/j.cell.2021.01.025

Reviewer Reports on the First Revision:

Referees' comments:

Referee #1 (Remarks to the Author):

In my initial review of the manuscript by Fu et.al. I was overall positive, and suggested small comments to address before the paper would be suitable for publication. I am satisfied with the responses to these points I made previously.

In reading the comments of the other reviewers, I see there is a robust discussion of other aspects of the manuscript, and which seems worth considering, but which I am not qualified to speak about.

Referee #2 (Remarks to the Author):

Unfortunately, the authors have largely addressed my comments with critiques of them and of the published literature, rather than using further experimental data to directly address key issues that were raised. These key issues are summarized as follows:

Significance: The idea of karyopherin imitation by the capsid isn't new, and the absence of key citations that have shown this really weakens the manuscript. A notable example is the recent publication in your affiliated journal that delved into many of the HIV-related subjects these papers touch on, making much of their content less novel: Xue, 2023, 37355754 - available on BioRxiv since 2021. Specifically, Xue et al. titled their work "the HIV-1 capsid core is an opportunistic nuclear import receptor", exactly the theme repeated in this current manuscript. Many studies, including Kane et al., 2018, Matreyek et al., 2013, Price et al., 2014, Bhattacharya et al., 2014, Bichel et al., 2013, and Lin et al., 2013 have explored the capsid's interaction with nucleoporins like FG repeats, and show that the key players for capsid entry into the NPC are Nup358, Nup214, Nup62, Pom121 and Nup153. Thus the Karyopherin-like behaviors of CA isn't a revolutionary idea, with foundational papers on the structures of CA binding to FG repeats dating back to 2014, and other significant structural and biochemical exploration of these in e.g. Shen, 2023, 36943880, and Shen, 2023, 36807645. The field's main question - whether the capsid moves completely through the NPC in vivo - also remains unanswered.

Significance and relevance of Nup98 analog condensates to HIV import: Another serious issue is the emphasis on the Nup98 FG repeat type, and its homologs and artificial variants. The repeat sequence Nup98 analog is an excellent biophysical model for a condensate, and the Gorlich group have extensively characterized those physical characteristics. But as indeed stated by them, the molecular interactions with NTRs and the nature of facilitated diffusion (as opposed to recruitment within condensates) is far from clear (Najbauer et al., 2022, 35314668). So, while crucial for regular transport, Nup98 has been repeatedly shown not to be among the most pivotal nucleoporins for HIV's NPC entry, and there's little direct evidence that Nup98 is a key player in capsid import. As indicated also above, multiple papers indicate that other FG Nups and while potentially involved (e.g. Di Nunzio et al., 2023) it is not a key player (e.g. Xue et al., 2023, Dharan et al., 2020, Ao et al.,

2012, Bichel et al., 2013, Matreyek et al., 2011, Price et al., 2014, Kane et al., 2018, Buffone et al., 2018). To address these published data, the authors should present evidence - not critiques of others' work – that Nup98 is crucial for HIV import in vivo. This manuscript does touch upon capsid binding to multiple FG Nups in vitro, but given the depth of existing literature, it's not groundbreaking and doesn't explore the capsid's capacity to navigate media involving these critical nucleoporins in a realistic NPC model, unlike e.g. Shen, 2023 36807645.

Nup98 and analog condensates as an appropriate model for HIV NPC entry: The manuscript also focuses on the capsid's interactions with FG Nup condensates. While this model effectively captures FG binders, its role in NPC structure and function in vivo is contentious. To assume that in vitro interactions with condensates equate to in vivo NPC processes is a far-reaching claim, especially given a recent paper from Nature that shows the contrary in vivo (Yu, 2023, PMID 37100914). It's unclear what value the Nup98 etc. condensate model adds, especially when compared to e.g. entry of HIV into the NPC origami model of Shen et al., 2023, the work of Yu et al., 2023, and Hudait and Voth's 2023 work, which offers a detailed look at HIV capsid transport into the NPC and the latter indicates that condensate formation can actually hinder capsid entry.

It is clear in actual NPCs in vivo Nup98 does not exist in isolation and other FG Nups – and association with non-FG regions of Nups - are necessary for capsid entry into the NPC. This manuscript begins by giving the impression that they believe this Nup98 analog condensate state is representative of the state of all the FG Nups in vivo. If this is not the case, they should make this far clearer in their text; if so, then more evidence for that being the case is needed. In this regard, the title of the paper is clearly a misrepresentation, as 'to traverse the permeability barrier;' is not demonstrated in the manuscript, and is only inferable from a hypothesis that the absorption of the capsid to the condensate is equivalent to facilitated transport of the capsid, which is clearly a stretch. This is a faulty analogy and is a consistent theme of the alleged role of condensate as a model for NPC mechanisms. Specifically, the NPC provides a facilitated diffusion mechanism while the condensate states provide only recruitment of factors, and their facilitated release is not (or very weakly) demonstrated.

The manuscripts should be less biased, and clearly state that the field is not settled as to the state that FG Nups form in the NPC in vivo. Giving any other impression, and dismissing the work of numerous other groups, is very unrepresentative of the active state of the field.

Probably, this comment...

Agreed, I intended to refer to the experiment as described.

This is an incomplete and inexact account of our findings...

Then what is needed is a more exact description in the main text. I see no issue with the word "posit" in the sense of "propose" or "put forth", based on their findings.

Our introduction gave an accurate account...

I still disagree, as once again the authors' point is based on the assumption that the condensate is an accurate representation of capsid entry via interaction with key sets of different FG nucleoporins in

vivo, which this manuscript still has not shown. As stated above, the demonstration of interactions with condensates may be an interesting biophysical model, but does not significantly add to the biological relevance already revealed by the studies of others. For example, the condensate could be formed in a constrained device such as in Shen, 2023 or other published model systems, show that in such a system the protein remains as a condensate in that nanoscopic environment, and show that the capsid traverses along a concentration gradient entirely through the condensate from one side to the other.

These points do not conflict with our main story...

The question pertains to the relative importance of the interactions with the phase separated form of Nup98 or its analogs as shown in this manuscript, and the fact that the field has instead in various papers (reviewed in e.g. Shen et al., 2021) shown that it is other Nups, including other FG Nups not shown here to form such a selective phase for capsids, and other accessory factors that play the most important roles in HIV access to the NPC. This was raised by the arbitrating reviewer, who also cited Xue et al., 2023. The authors are dismissive of this work – but it is a publication in a sister journal of Nature, that clearly shows experimental evidence of only a minor role for Nup98 in HIV accessing the NPC, but similar knockdowns showed an important role for the FG repeat containing Nup153 and Pom121 (as well as Nup358), indicating the main FG repeat interacting Nups in vivo might be proteins other than Nup98. This reviewer and this paper thus underscore my concerns. To address this, the authors would need to show in vivo experimental evidence that Nup98 is specifically required (as they point out, its general requirement is a given since it is lethal upon removal) for HIV import in vivo. Secondly, again Nup98 does indeed undergo phase separation in solution, but this is not a generally observed phenomenon for all FG Nups.

A comprehensive model of HIV-1 infection also...

I disagree with the authors that they have demonstrated how the capsid overcomes the FG phase-based permeability barrier in several respects. First, the phase-base of the barrier remains to be established, and is indeed not supported in several aspects including Kozai, 2023, 37066338 with a comparison study of intact NPCs and condensates, and in other papers I have cited above. Secondly, the prior suggestion of NTR-like interactions from studies quoted above have already provided abundant evidence that capsid is karyopherin-like and can thereby overcome the NPC barrier by similar mechanism.

With all due respect to the reviewer, everybody is entitled to their own opinion, but not to their own facts...

It is unfortunately the authors who are not providing all the facts. The Yu et al., 2023 paper which the authors claim “agree with us that Nup98 condensates indeed show very similar transport selectivity as NPC” with Nup98 condensates in fact says: “Despite having similar permeability barrier properties as the intact NPC, the bulk condensate formed from phase separating NUP98 is an incomplete approximation of the actual permeability barrier, the materials properties of which are modulated by the anchoring of a distinct number of FG-NUPs with 3D precision on a half-toroidal

NPC scaffold". While there are references that use FG condensates as a model of transport, there are many references that do otherwise and question their relevance to transport in vivo, and just in recent reviews, the fact that this aspect of the transport mechanism remains unresolved and controversial is expounded e.g. in Huang, 2020, 32794558; Hoogenboom, 2021 35892075; Kalita, 2022, 35089308; Cowburn, 2023 37099395; Zheng, 2023, 36757893.

On re-reading of the Lim paper, it is not clear (nor do the authors state) how it "grossly misrepresents the FG phase literature". Moreover, it makes exactly the same point as the Yu et al. paper and the point I'm making here – namely, that while "FG condensates do indeed show NPC-like properties" they don't necessarily faithfully represent how the entire population of FG Nups behave in the NPC in situ. The bone of contention here - and throughout the whole manuscript – is thus how faithfully these condensates mimic the in vivo situation. The authors state in this rebuttal, their intention is to show that "the assay shows NTRness... by the standards of the assay, HIV-1 capsids are indeed NTR-like". However, I've gained the impression throughout this manuscript that instead the authors are implying that they are faithfully mimicking the state of FG Nups in vivo. If they are not doing so – that their condensates are merely an assay for bulk cooperative binding of FG repeats to CA in capsids as a prerequisite for transport - then statements that are unambiguously to this effect should be made clearly in the manuscript in the introduction and conclusions.

We do indeed demonstrate that...

The new data do clarify this point. Indeed, the capsid inserts into the NPC without other factors. But the authors are concatenating two separate pieces of data - insertion into the NPC, and absorption into a condensate - into assuming they are mechanistically one and the same process. But, as stated above, this has not been proven to be true. Experiments along the lines suggested above could directly address this.

This statement inaccurately reflects...

The comment was indeed directed mainly to the Dickson et al. manuscript. However, as the authors also show data pertaining to condensates made out of the orthologs human Nup98, yeast Nup116, trypanosome Nup158, and SLFG and FSFG repeats (I assume derived from the engineered GLFG one), could they state in the Methods or figure legends the timing of each experiment and what is their estimate for the condensates to start showing deleterious gelification and amyloid formation (in the cases where this could be an issue)? I was questioning the behavior across the entire time course of "aging" i.e. probing what different states ""aging", gelation, and formation of potentially aberrant beta-structures" are being formed at the time points given. This pertains to our question, "How would the "aging" of the condensates influence the different experiments presented in these manuscripts"? I am just asking for controls that for these experiments with the capsids, are the Nup98 condensates changing state during the time course of the experiments? I understand that the artificial Nup98 analog is engineered to try and avoid these issues, but that does not carry over onto the other natural repeats.

This statement inaccurately describes our findings...

No, it doesn't – and yes, the wording was similar because I was making a similar point - and fortunately, for this point the authors performed an experiment to directly address our query. Gratifyingly, the experiment does indeed address our point, showing “that capsid entry into the GLFG phase resists competition by ~physiological concentrations of traditional NTRs”.

Not all the FG particles are in the same plane...

Rim-staining: this might again be a comment for the parallel paper

No, this isn't a comment for the accompanying paper. The explanation provided in response seems reasonable – but could the authors provide serial confocal slices of a representative set of larger and smaller condensates to show the distribution of CA-particles in them?

The Lim paper does not appear to “generalize from Nup100 to other FG domains”, as e.g. they examine multiple FG domains and their propensity to form condensates *in vivo*. Instead, it's concerning that the authors no longer consider Nup100 a suitable model for FG condensates, as they have used it prominently as such in the past e.g. Schmidt & Gorlich, 2015 25562883, where its selective behavior is compared favorably with its homolog Nup98. This shifting ground underscores my issues with Nup98's use here as a new and better model for NPC functionality - the authors have assumed, as they did in this previous paper, that Nup98 is a good mimic of transport, even though now Nup100 - once also a good model - is no longer considered so by them. Moreover, the model system used by the authors here are variations on just one nup, Nup98, that gives them exactly the properties they want - *in vitro* liquid condensates - but not necessarily those found *in vivo* in the NPC, or formed by other FG nucleoporins, including those known to be involved in HIV entry into the NPC - see points above.

Referee #3 (Remarks to the Author):

The field of HIV nuclear import has been highly contentious despite decades of research and multiple reports, many of these in the highest impact journals. Seminal work nearly 20 years ago from Emerman and colleagues (2004 *J Virol*) first highlighted HIV capsid as the key mediator, debunking at the time earlier work focused on Vpr, matrix, integrase, and “the central DNA flap”. The work in this paper and the accompanying Dickson manuscript now provide the biochemical and biological basis for capsid-mediated HIV nuclear import. These papers together represent a transformative advance for the field and should be published in *Nature*. It is unjustified to suggest that these studies advance the field incrementally.

Suggestions for improvement:

1) The work moreover opens up new avenues for research. The NTR-mimic concept, which is novel (despite the overclaim of a recent *Nat Commun* title), indicates that the core navigates the pore via multiple low affinity yet high valency FG interactions. The authors at times however seemingly over characterize all known capsid-FG interactions into this one basket. Page 6, 6 lines up from bottom “is not adapted to a specific FG repeat sequence”. While this may apply for most of the novel interactions studied here (Fig 2a), prior work has indicated otherwise for Nup153: although Nup153

is an FG Nup with a bona fide FG domain, ref 26 and PMID 36943880 highlighted capsid binding to but a single FG. The “specialized FG motifs” description page 4, which is apt, should carry over to page 6 discussion.

2) Ref 39 concluded that HIV hexamers but not pentamers harbor functional FG binding sites. Yet, the partition coefficient of your spheres, which are predominantly if not exclusively pentameric, approaches that of CLPs (Fig 4b, c). Please comment on this apparent contradiction. Might this depend on the analyzed FG?

3) First full paragraph page 4 (We reasoned that...) does not do prior works justice. Ref 33 and 26 first showed CPSF6 and Nup153 FG peptides bind the CA N-terminal domain. Refs 22-24 then showed these bind with greater affinity to the hexamer. PMID 36202818 further showed higher affinity binding of FG-containing fragments to mature capsid lattices over hexamers. “through hydrophobic binding pockets created by hexamerization” dismisses the initial studies and overlooks the 2022 Nat Commun findings.

4) Same paragraph line 4, since your citations now include a 2023 paper, “earlier” does not apply. These citations should include PMID 30084827.

5) I apologize for not requesting this earlier, but SDS-PAGE images of the various Nup peptides and control proteins used in BLI assays (Fig 2 and ED Fig 2) should be shown.

6) It seems possible to calculate apparent Kds from BLI data. Please consider adding this info.

7) Added ED Fig 5 is elegant and ups biological credence. A non-binding CLP/sphere is a serious control to consider. Might you have data for N57A structures?

8) For completeness, Fig 3d should be expanded to include capsomeres shown in panel b.

9) Page 4 “pointing to the capsid interior”. Is there data to show this? (or publication to cite?).

Minor:

10) ED Fig 4 is cited in text prior to ED Fig 3 (page 6). Please renumber the Figs to reflect the order in which they are cited in main text.

11) Fig 4 Title “Giant” is overly dramatic and should be deleted. These are normal HIV size.

Author Rebuttals to First Revision:

Point-by-point response (Reviewers' queries are repeated in blue in front of each of our answers)

Referee #1 (Remarks to the Author):

In my initial review of the manuscript by Fu et.al. I was overall positive, and suggested small comments to address before the paper would be suitable for publication. I am satisfied with the responses to these points I made previously.

We thank the Referee for their support for publication of the study.

Referee #2 (Remarks to the Author):

1. Unfortunately, the authors have largely addressed my comments with critiques of them and of the published literature, rather than using further experimental data to directly address key issues that were raised.

Indeed, we have addressed all constructive comments and where appropriate also with additional experimental. These additional data included, e.g., the demonstration of a complete NPC passage of capsid spheres in the oocyte system, as well as the documentation of a capsid mutation that abrogates FG-binding, FG phase partitioning as well complete NPC passage.

2. These key issues are summarized as follows:

Significance: The idea of karyopherin imitation by the capsid isn't new, and the absence of key citations that have shown this really weakens the manuscript. A notable example is the recent publication in your affiliated journal that delved into many of the HIV-related subjects these papers touch on, making much of their content less novel: Xue, 2023, 37355754 - available on BioRxiv since 2021. Specifically, Xue et al. titled their work "the HIV-1 capsid core is an opportunistic nuclear import receptor", exactly the theme repeated in this current manuscript.

We had answered this point already in our answer to the arbitrating reviewer #4, and reviewer #3 (see below), the HIV expert, also explicitly points out that the title used by Xue et al. is an overstatement.

Xue et al. does not provide any evidence that would possibly support the claim of their title. They show that the knockdown of two FG Nups (Pom121, Nup153, both being involved in NPC biogenesis) reduced infection by HIV-1, while a knockdown of other FG Nups (Nups 54, 58, 62, 98, 214) had no effect (see figure 1a of their paper). The experiment that is still closest to the claim is a pulldown with overexpressed Pom121 as a bait that showed some capsid binding. However, since the pulldown was from a complete cell lysate with all cellular NTRs being also present, it cannot distinguish between a direct FG-capsid interaction (as expected for an autonomous NTR) and an interaction bridged by cellular NTRs (as expected for a normal cargo).

The main part of this paper was about Nup35 with the authors apparently believing that Nup35 is an FG Nup. But, as this reviewer surely knows as well, Nup35 is not an FG Nup (as explained in our detailed answer to the arbitrating reviewer 4).

3. Many studies, including Kane et al., 2018, Matreyek et al., 2013, Price et al., 2014, Bhattacharya et al., 2014, Bichel et al., 2013, and Lin et al., 2013 have explored the capsid's interaction with nucleoporins like FG repeats, and show that the key players for capsid entry into the NPC are Nup358, Nup214, Nup62, Pom121 and Nup153. Thus the Karyopherin-like behaviors of CA isn't a revolutionary idea, with foundational papers on the structures of CA binding to FG repeats dating back to 2014, and other significant structural and biochemical exploration of these in e.g. Shen, 2023, 36943880, and Shen, 2023, 36807645.

We show that the capsid (approximated as 40 nm capsid spheres or 100 nm CLPs) is drawn into an FG phase/barrier (to a partition coefficient of ≥ 500), while much smaller inert macromolecules such as mCherry remain strictly excluded (to a partition coefficient of ≤ 0.05). This highly selective barrier crossing is a different quality from the mere binding by FG repeats reported in the studies cited above. This is the key point that reviewer #2 seems to be missing.

4. The field's main question – whether the capsid moves completely through the NPC in vivo – also remains unanswered.

We addressed this question in Extended Data Figure 5, showing that capsid spheres microinjected into the cytoplasm of (living) mouse oocytes passed through NPCs and reached the nuclear interior, while the far smaller injection control remained cytoplasmic, indicating that the capsids had crossed an intact NPC barrier. We have now added another specificity control showing that the well-known N57A mutation not only impedes the partitioning of the capsid into the GLFG phase but also blocks its passage through the oocyte NPCs (new fig. 4 and ED fig. 6). As the mutation also impedes infection, we have a direct genetic link between these three readouts.

As pointed out by reviewer #3 in their previous evaluation, there is good virological evidence for the capsid completing NPC passage and uncoating inside nuclei. We had adjusted our text to accommodate this prior knowledge.

5. Significance and relevance of Nup98 analog condensates to HIV import: Another serious issue is the emphasis on the Nup98 FG repeat type, and its homologs and artificial variants. The repeat sequence Nup98 analog is an excellent biophysical model for a condensate, and the Gorlich group have extensively characterized those physical characteristics. But as indeed stated by them, the molecular interactions with NTRs and the nature of facilitated diffusion (as opposed to recruitment within condensates) is far from clear (Najbauer et al., 2022, 35314668). So, while crucial for regular transport, Nup98 has been repeatedly shown not to be among the most pivotal nucleoporins for HIV's NPC entry, and there's little direct evidence that Nup98 is a key player in capsid import. As indicated also above, multiple papers indicate that other FG Nups and while potentially involved (e.g. Di Nunzio et al., 2023) it is not a key player (e.g. Xue et al., 2023, Dharan et al., 2020, Ao et al., 2012, Bichel et al., 2013, Matreyek et al., 2011, Price et al., 2014, Kane et al., 2018, Buffone et al., 2018). To address these published data, the authors should present evidence – not critiques of others' work – that Nup98 is crucial for HIV import in vivo. This manuscript does touch upon capsid binding to multiple FG Nups in vitro, but given the depth of existing literature, it's not groundbreaking and doesn't explore the capsid's capacity to navigate media involving these critical nucleoporins in a realistic NPC model, unlike e.g. Shen, 2023 36807645.

We have previously demonstrated that a cohesively interacting Nup98 FG domain is required for building a permeability barrier in NPCs (Hülsmann *et al.*, 2012). NPCs that lack Nup98-like FG repeats are non-selectively permeable. It simply is the wrong concept to expect the most critical barrier-forming FG Nup to be a host factor that would *promote* infection. It is a barrier to infection, overcome by the NTR-properties of the HIV-1 capsid.

Many of the previous studies drew conclusions from isolated, biochemical examination of CA-monomers, -hexamers, -tubes, where Nup98-derived FG-segments register as weaker binders than, say, the specific Nup153 or CPSF6 FG-peptides for which co-crystal structures exist. But it is one of the major thrusts of this and the accompanying Dickson et al. study, that we both now show that for nuclear transport we need to consider the entire FG-phase and CA-spheres. Under these conditions, where multivalency plays a key role, the conclusions are noticeably different, and that is where we believe our studies significantly advance the field.

6. Nup98 and analog condensates as an appropriate model for HIV NPC entry: The manuscript also focuses on the capsid's interactions with FG Nup condensates. While this model effectively captures FG binders, its role in NPC structure and function in vivo is contentious. To assume that in vitro interactions with condensates equate to in vivo NPC processes is a far-reaching claim, especially given a recent paper from Nature that shows the contrary in vivo (Yu, 2023, PMID 37100914). It's unclear what value the Nup98 etc. condensate model adds, especially when compared to e.g. entry of HIV into the NPC origami model of Shen et al., 2023, the work of Yu et al., 2023, and Hudait and Voth's 2023 work, which offers a detailed look at HIV capsid transport into the NPC and the latter indicates that condensate formation can actually hinder capsid entry.

Every model has its limitations. Our entire claim merely is that the FG-phase mimics the NPC transport channel in key parameters that can be described as 'NTRness' for viral cores. Not more, not less. We have modified the text to address these limitations more clearly.

7. It is clear in actual NPCs in vivo Nup98 does not exist in isolation and other FG Nups – and association with non-FG regions of Nups - are necessary for capsid entry into the NPC. This manuscript begins by giving the impression that they believe this Nup98 analog condensate state is representative of the state of all the FG Nups in vivo. If this is not the case, they should make this far clearer in their text; if so, then more evidence for that being the case is needed. In this regard, the title of the paper is clearly a misrepresentation, as ‘to traverse the permeability barrier;’ is not demonstrated in the manuscript, and is only inferable from a hypothesis that the absorption of the capsid to the condensate is equivalent to facilitated transport of the capsid, which is clearly a stretch. This is a faulty analogy and is a consistent theme of the alleged role of condensate as a model for NPC mechanisms. Specifically, the NPC provides a facilitated diffusion mechanism while the condensate states provide only recruitment of factors, and their facilitated release is not (or very weakly) demonstrated. The manuscripts should be less biased, and clearly state that the field is not settled as to the state that FG Nups form in the NPC in vivo. Giving any other impression, and dismissing the work of numerous other groups, is very unrepresentative of the active state of the field.

The scope of our present manuscript was not to experimentally define the state of all FG domains in an NPC *in vivo*, nor to review the entire literature on this topic. Instead, we have addressed the question of how the HIV-1 capsid is able to overcome the permeability barrier of NPCs.

There is little doubt that the FG domain of Nup98 is the most barrier-critical one. It provides the largest share of FG mass and its removal leads to non-selectively permeable NPCs. Independent of any model assumptions, the (Nup98) FG phase system is the so far only experimental system that can recapitulate the transport selectivity of NPCs, and it does so in remarkable detail. The FG phase partition coefficients of mobile species are good predictors of their NPC passage rates, and this over 4 orders of magnitude (!) of rates and partition coefficients (Frey et al., 2018). The phase system faithfully recapitulates the NTR-dependence of cargo transport, be it importin-, exportin-, or NTF2 dependent (Ng et al., 2021). It faithfully recapitulates the coupling to the RanGTPase system as well as the multi-NTR requirement for larger cargoes (Schmidt and Görlich, 2015). Since the capsid behaves like an NTR in this well-studied system, it is also valid to conclude that the capsid has NTR properties. This conclusion is further supported by our observations of a highly efficient and NTR-independent capsid targeting to NPCs, the observed NPC passage in the mouse oocyte system and the fact that the N57A mutation not only disrupts capsid-FG interactions but also capsid partitioning in the Nup98 FG phase and the capsid passage through NPCs. It is actually hard to think of a more coherent and comprehensive chain of evidence.

Of course, there are other FG Nups in animal NPCs. We have previously reported that they form similar FG phases with similar transport selectivity as a Nup98 phase (Labokha *et al.*, 2013). This was true for the FG domains of Nup358, Nup153, Nup214 and the Nup54-58-62 complex. All of these form FG hydrogels that repel inert macromolecules but allow entry of NTRs and NTR-cargo complexes, and we would expect them to allow entry of the HIV-1 capsid as well (given that capsid hexamers bind all the FG domains tested and that capsid spheres enter condensed FG phases with a range of FG motifs). These other FG hydrogels appeared to be less restrictive than the Nup98 FG phase, so if there is a difference, we would expect the capsid to penetrate, for example, Nup214 or Nup153 FG hydrogels more readily than a Nup98 phase. We will report on this in follow-up studies.

This reviewer re-iterated the same point over and over again, with minor variations. To minimize redundancies, we answer this point in more detail here. The reviewer listed below several references and claimed that we should have cited them. Most of them entertain the entropic brush model for NPC selectivity (or variations thereof). The brush model considers cohesive (barrier-forming) interactions between FG repeats as irrelevant. In contrast, the selective phase model (and its variants) assume that multivalent, reversible, cohesive interactions create a sieve-like barrier, into which NTRs can selectively melt because their FG-binding can disengage the reversible inter-FG contacts.

How to distinguish between these models? The most stringent test would be to replace the cohesively interacting FG repeat domains in real NPCs for non-cohesive ones, and then test the barrier properties. If cohesive interaction were irrelevant, then nothing should change. We performed an entire series of experiments along these lines and observed that exchanging the highly cohesive Nup98 FG repeats for other highly cohesive ones retained function. Any exchange for non-cohesive or only partially cohesive FG repeat domains, however, led to NPCs of non-selective permeability that failed to exclude, e.g., dextrans and failed to retained imported cargoes inside nuclei (Hülsmann *et al.*, 2012). We therefore consider the cohesive FG interactions as essential and the brush model as obsolete. This has been thoroughly peer-reviewed, it is published, and highly cited.

8. Probably, this comment...

Agreed, I intended to refer to the experiment as described.

This is an incomplete and inexact account of our findings...

Then what is needed is a more exact description in the main text. I see no issue with the word “posit” in the sense of “propose” or “put forth”, based on their findings.

Our introduction gave an accurate account...

I still disagree, as once again the authors’ point is based on the assumption that the condensate is an accurate representation of capsid entry via interaction with key sets of different FG nucleoporins *in vivo*, which this manuscript still has not shown. As stated above, the demonstration of interactions with condensates may be an interesting biophysical model, but does not significantly add to the biological relevance already revealed by the studies of others. For example, the condensate could be formed in a constrained device such as in Shen, 2023 or other published model systems, show that in such a system the protein remains as a condensate in that nanoscopic environment, and show that the capsid traverses along a concentration gradient entirely through the condensate from one side to the other.

Shen 2023 comes to the conclusion of an affinity model based on their observations, which are not consistent with what we observe, nor are they consistent with previous work. It simply illustrates that the issue of NPC transport of HIV-1 capsids is not settled, contrary to the referee’s opinion. We revisited our manuscript and made sure that we do not overstate the relevance of our observations.

9. These points do not conflict with our main story...

The question pertains to the relative importance of the interactions with the phase separated form of Nup98 or its analogs as shown in this manuscript, and the fact that the field has instead in various papers (reviewed in e.g. Shen et al., 2021) shown that it is other Nups, including other FG Nups not shown here to form such a selective phase for capsids, and other accessory factors that play the most important roles in HIV access to the NPC. This was raised by the arbitrating reviewer, who also cited Xue et al., 2023. The authors are dismissive of this work – but it is a publication in a sister journal of Nature, that clearly shows experimental evidence of only a minor role for Nup98 in HIV accessing the NPC, but similar knockdowns showed an important role for the FG repeat containing Nup153 and Pom121 (as well as Nup358), indicating the main FG repeat interacting Nups *in vivo* might be proteins other than Nup98. This reviewer and this paper thus underscore my concerns. To address this, the authors would need to show *in vivo* experimental evidence that Nup98 is specifically required (as they point out, its general requirement is a given since it is lethal upon removal) for HIV import *in vivo*. Secondly, again Nup98 does indeed undergo phase separation in solution, but this is not a generally observed phenomenon for all FG Nups.

Again, we are not making the argument that FG-Nups other than Nup98 are not important. We, in fact, show that FG-Nups, in general, dynamically bind to CA. Our study is not about an argument about which FG-Nup is most important for HIV-1 import. Given the multifaceted nature of the NPC, this is actually a very difficult question to answer. The often mentioned Yu et al. 2023 study serves as a wonderful example. For instance, that manuscript heavily focuses on the role of the non-FG-Nup35 on HIV-1 nuclear transport, presumably because Nup35 manipulations have an effect on FG-Nup anchorage to the NPC-channel and thus likely influence the transport process indirectly rather than directly.

10. A comprehensive model of HIV-1 infection also...I disagree with the authors that they have demonstrated how the capsid overcomes the FG phase-based permeability barrier in several respects. First, the phase-base of the barrier remains to be established, and is indeed not supported in several aspects including Kozai, 2023, 37066338 with a comparison study of intact NPCs and condensates, and in other papers I have cited above. Secondly, the prior suggestion of NTR-like interactions from studies quoted above have already provided abundant evidence that capsid is karyopherin-like and can thereby overcome the NPC barrier by similar mechanism.

This comment is essentially covered by the responses above.

11. With all due respect to the reviewer, everybody is entitled to their own opinion, but not to their own facts...It is unfortunately the authors who are not providing all the facts. The Yu et al., 2023 paper which the authors claim “agree with us that Nup98 condensates indeed show very similar transport selectivity as NPC” with Nup98 condensates in fact says: “Despite having similar permeability barrier properties as the intact NPC, the bulk condensate formed from phase separating NUP98 is an incomplete approximation of the actual permeability barrier, the materials properties of which are modulated by the anchoring of a distinct number of FG-NUPs with 3D precision on a half-toroidal NPC scaffold”. While there are references that use FG condensates as a model of transport, there are many references that do otherwise and question their relevance to transport in vivo, and just in recent reviews, the fact that this aspect of the transport mechanism remains unresolved and controversial is expounded e.g. in Huang, 2020, 32794558; Hoogenboom, 2021 35892075; Kalita, 2022, 35089308; Cowburn, 2023 37099395; Zheng, 2023, 36757893.

We believe that we have sufficiently answered this concern in response to comment 7 above.

12. On re-reading of the Lim paper, it is not clear (nor do the authors state) how it “grossly misrepresents the FG phase literature”. Moreover, it makes exactly the same point as the Yu et al. paper and the point I’m making here – namely, that while “FG condensates do indeed show NPC-like properties” they don’t necessarily faithfully represent how the entire population of FG Nups behave in the NPC in situ. The bone of contention here – and throughout the whole manuscript – is thus how faithfully these condensates mimic the in vivo situation. The authors state in this rebuttal, their intention is to show that “the assay shows NTRness... by the standards of the assay, HIV-1 capsids are indeed NTR-like”. However, I’ve gained the impression throughout this manuscript that instead the authors are implying that they are faithfully mimicking the state of FG Nups in vivo. If they are not doing so – that their condensates are merely an assay for bulk cooperative binding of FG repeats to CA in capsids as a prerequisite for transport – then statements that are unambiguously to this effect should be made clearly in the manuscript in the introduction and conclusions.

This comment is essentially covered by the responses above.

13. We do indeed demonstrate that...

The new data do clarify this point. Indeed, the capsid inserts into the NPC without other factors. But the authors are concatenating two separate pieces of data – insertion into the NPC, and absorption into a condensate – into assuming they are mechanistically one and the same process. But, as stated above, this has not been proven to be true. Experiments along the lines suggested above could directly address this.

Our conclusion of NTR-like properties of the HIV capsid rests on way more than just two isolated observations. There is a large body of prior knowledge about NTRs (importin β -related, NTF2-related, engineered ones), we know that cohesive (FG phase-forming) inter FG interactions are essential for the NPC barrier (Hülsmann *et al.*, 2012), we know that NPC passage rates of a mobile species correlates very well with its partitioning into a Nup98 FG phase (and this over 4 orders of magnitudes of rates and partition coefficients; Frey *et al.*, 2018), we know that the phase separation propensity of Nup98 FG domains is strictly conserved across all eukaryotic clades, etc. We show here not only that the capsids target NPCs with extraordinary efficiency and specificity but also that this happens in an importin-independent manner, that the capsid partition very efficiently into highly transport-selective FG phase as otherwise only NTRs do. We demonstrate rapid and complete NPC passage of capsid spheres and that this passage is blocked by a mutation that interferes with direct FG interactions.

14. This statement inaccurately reflects...

The comment was indeed directed mainly to the Dickson et al. manuscript. However, as the authors also show data pertaining to condensates made out of the orthologs human Nup98, yeast Nup116, trypanosome Nup158, and SLFG and FSFG repeats (I assume derived from the engineered GLFG one), could they state in the Methods or figure legends the timing of each experiment and what is their estimate for the condensates to start showing deleterious gelification and amyloid formation (in the cases where this could be an issue)? I was questioning the behavior across the entire time course of “aging” i.e. probing what different states “aging”, gelation, and formation of potentially aberrant beta-structures” are being formed at the time points given. This pertains to our question, “How would the “aging” of the condensates influence the different experiments

presented in these manuscripts”? I am just asking for controls that for these experiments with the capsids, are the Nup98 condensates changing state during the time course of the experiments? I understand that the artificial Nup98 analog is engineered to try and avoid these issues, but that does not carry over onto the other natural repeats.

The GLFG phase used here appears to be in a thermodynamically stable state. Phase separation occurs instantaneously upon dilution from the FG domain stock (that is kept initially non-interacting by 2-4 M guanidinium hydrochloride), and after that, there is no change of properties over time. In the case of this perfectly repeated GLFG52x12 domain, we followed a time course over months, using NMR as a readout. No signs for cross β -structures detected – consistent with the lack of Thioflavin-T staining.

N/Q-rich FG domains (Nup100 or Nup116) stain positive with Thioflavin-T and by this criterion they contain amyloid-like cross β -structures. The thioflavin signal increases over time (as we reported before Ng *et al.*, 2021). However, the protein density and the partition coefficients for mobile species/ clients remains constant for at least a week.

There are some technical details to pay attention to. For example, a drying of the sample will change the microscopic appearance. Likewise, FG domains containing cysteines (all GLEBS domain-containing ones do) may form disulfide bonds over time (i.e., within days), unless a reducing agent like TCEP, GSH or DTT is present. This is one reason why our model sequence (perf. GLFG52x12) is cysteine-free, but this is a rather trivial technicality.

The timing of the experiment was already described in the first submitted version. We show 1 hour time points of substrate entry into pre-formed FG phases, following a ~5 minutes incubation to complete FG phase assembly (here, seconds would be sufficient as the outcome does not change with shorter pre-incubations).

15. This statement inaccurately describes our findings...

No, it doesn't – and yes, the wording was similar because I was making a similar point - and fortunately, for this point the authors performed an experiment to directly address our query. Gratifyingly, the experiment does indeed address our point, showing “that capsid entry into the GLFG phase resists competition by ~physiological concentrations of traditional NTRs”.

Thank you.

16. Not all the FG particles are in the same plane... Rim-staining: this might again be a comment for the parallel paper

No, this isn't a comment for the accompanying paper. The explanation provided in response seems reasonable – but could the authors provide serial confocal slices of a representative set of larger and smaller condensates to show the distribution of CA-particles in them?

For z-stacks, please see below. It is a repetition of fig.3b, comparing the partitioning of sinGFP4a-labelled capsomers (which remain arrested at the FG phase surface) and the identically labelled 40 nm spheres (which fully enter the FG phase). We also provide series of yz-scans of the same samples, which give a complementary visual impression.

sinGFP4a-labeled Capsomeres

X-Y plane: Z stack (from top to bottom)

sinGFP4a-labeled 40 nm spheres

X-Y plane: Z stack (from top to bottom)

For Editors' & Reviewers' information only.

sinGFP4a-labeled Capsomeres

X-Z plane: Y stack

sinGFP4a-labeled 40 nm spheres

X-Z plane: Y stack

For Editors' & Reviewers' information only.

17. The Lim paper does not appear to “generalize from Nup100 to other FG domains”, as e.g. they examine multiple FG domains and their propensity to form condensates *in vivo*. Instead, it’s concerning that the authors no longer consider Nup100 a suitable model for FG condensates, as they have used it prominently as such in the past e.g. Schmidt & Görlich, 2015 25562883, where its selective behavior is compared favorably with its homolog Nup98. This shifting ground underscores my issues with Nup98’s use here as a new and better model for NPC functionality - the authors have assumed, as they did in this previous paper, that Nup98 is a good mimic of transport, even though now Nup100 - once also a good model - is no longer considered so by them. Moreover, the model system used by the authors here are variations on just one nup, Nup98, that gives them exactly the properties they want - *in vitro* liquid condensates - but not necessarily those found *in vivo* in the NPC, or formed by other FG nucleoporins, including those known to be involved in HIV entry into the NPC - see points above.

The reviewer is taking our answer out of context. We responded to the reviewer's assertion that we would be looking at aging effects. Aging effects are relevant when a condensate becomes an amyloid. This might only be an issue for the Nup100 FG domain, which is very N/Q-rich and in that related to the yeast prion Sup35 (Michelitsch and Weissman, 2000). However, it is an invalid extrapolation to assume amyloid-related aging effects for all other cohesive FG phases. For the GLFG phase used here, we can exclude any amyloid-forming propensity because it does not stain with Thioflavin-T (Schmidt and Görlich, 2015; Ng *et al.*, 2021) and because no cross- β structures are detectable by NMR, even after months of incubations (Najbauer *et al.*, 2022).

And yes, we were the first to publish that a Nup100 FG phase is highly transport-selective (Schmidt and Görlich, 2015). It allows entry of NTRs to very high partition coefficients, while excluding inert macromolecules. The observed selectivity is indeed NPC-like – even though this FG phase stains brightly with the amyloid-detecting probe Thioflavin-T. This is, however, not a topic of the current manuscript. The key conclusion of this manuscript is that the HIV-1 capsid behaves like an NTR, and this conclusion is coherently supported by orthogonal methods, including authentic NPCs of mammalian cells.

Referee #3 (Remarks to the Author):

The field of HIV nuclear import has been highly contentious despite decades of research and multiple reports, many of these in the highest impact journals. Seminal work nearly 20 years ago from Emerman and colleagues (2004 *J Virol*) first highlighted HIV capsid as the key mediator, debunking at the time earlier work focused on Vpr, matrix, integrase, and “the central DNA flap”. The work in this paper and the accompanying Dickson manuscript now provide the biochemical and biological basis for capsid-mediated HIV nuclear import. These papers together represent a transformative advance for the field and should be published in *Nature*. It is unjustified to suggest that these studies advance the field incrementally.

We thank the referee for this very positive evaluation of the two studies and for putting them into a perspective.

Suggestions for improvement:

1) The work moreover opens up new avenues for research. The NTR-mimic concept, which is novel (despite the overclaim of a recent *Nat Commun* title), indicates that the core navigates the pore via multiple low affinity yet high valency FG interactions. The authors at times however seemingly over characterize all known capsid-FG interactions into this one basket. Page 6, 6 lines up from bottom “is not adapted to a specific FG repeat sequence”. While this may apply for most of the novel interactions studied here (Fig 2a), prior work has indicated otherwise for Nup153: although Nup153 is an FG Nup with a bona fide FG domain, ref 26 and PMID 36943880 highlighted capsid binding to but a single FG. The “specialized FG motifs” description page 4, which is apt, should carry over to page 6 discussion.

We fully concur with the notion that certain FGs may indeed have a specialized role in capsid translocation, such as the well-studied Nup153 (aa1407-1423) FG-peptide. On page 6, the discussion is about traversing FG phases, which we show not to be dependent on a specific FG repeat sequence. We reworded the sentence to emphasize the context.

2) Ref 39 concluded that HIV hexamers but not pentamers harbor functional FG binding sites. Yet, the partition coefficient of your spheres, which are predominantly if not exclusively pentameric, approaches that of CLPs (Fig 4b, c). Please comment on this apparent contradiction. Might this depend on the analyzed FG?

For our 40 nm spheres we used the N21C/ A22C mutant, which was originally described as a T3 icosahedral assembly with 12 pentamers and 20 hexamers (Pornillos *et al.*, 2010). A later cryo-EM structure of the same assembly reported, however, a T4 icosahedral structure with 12 pentamers and 30 hexamers (Zhang *et al.*, 2018). So, these spheres are still dominated by hexamers. However, if only hexamers would bind FG motifs, then one should indeed expect a higher partition coefficient for CLPs than for the 40 nm spheres. Full agreement.

The reviewer is also correct that CA pentamers should not bind FG motifs in the same way as in the known hexamers-FG structures, as Schirra *et al.*, 2023 also concluded. However, this does not exclude that pentamers bind FGs; they might interact in a different way, which we think they probably do. We tested the G60A/ G61P double mutant, which assembles into 20 nm T1 icosahedral spheres with a pentamer-only arrangement (Schirra *et al.*, 2023; PDB 8EEP), and found that this assembly efficiently partitions into the Nup98 FG phase and also targets to HeLa NPCs [REDACTED]. This is beyond the scope of this study, but needs to be mentioned here to avoid the impression of implausibility within our dataset.

[REDACTED]

3) First full paragraph page 4 (We reasoned that...) does not do prior works justice. Ref 33 and 26 first showed CPSF6 and Nup153 FG peptides bind the CA N-terminal domain. Refs 22-24 then showed these bind with greater affinity to the hexamer. PMID 36202818 further showed higher affinity binding of FG-containing fragments to mature capsid lattices over hexamers. "through hydrophobic binding pockets created by hexamerization" dismisses the initial studies and overlooks the 2022 Nat Commun findings.

We thank the reviewer for pointing out these details and we apologize for our inaccuracy. The relevant sentence has been reworded to properly cite prior work: *"We reasoned that a solution to this conundrum might be related to the observation that the CA protein binds specialized FG motifs in the nuclear RNA polyadenylation factor CPSF6³³ and Nup153³⁴, enhanced through hydrophobic binding pockets created by hexamerization³⁵⁻³⁷, and further augmented when binding to mature HIV-1 lattices³⁸."*

4) Same paragraph line 4, since your citations now include a 2023 paper, "earlier" does not apply. These citations should include PMID 30084827.

We changed 'earlier' to 'other', PMID 30084827 is now included.

5) I apologize for not requesting this earlier, but SDS-PAGE images of the various Nup peptides and control proteins used in BLI assays (Fig 2 and ED Fig 2) should be shown.

ED Figs. 1 and 2 have been modified to include the requested SDS-PAGE controls of BLI-probes.

6) It seems possible to calculate apparent Kds from BLI data. Please consider adding this info.

Proper Kd measurements are straightforward only for monovalent analytes and ligands. In our setup, the analyte is the CA-hexamer with six binding sites. Most FG probes used as ligands are also themselves multivalent, with multiple FG repeats available for interaction with the CA-hexamer. Thus, the combination of multiple binding events and avidity effects preclude an accurate description by standard evaluation methods or by a simple Kd. These BLI data have primarily a qualitative value for detecting highly dynamic interactions.

7) Added ED Fig 5 is elegant and ups biological credence. A non-binding CLP/sphere is a serious control to consider. Might you have data for N57A structures?

We have now included these controls as new figure 4 and ED figure 6. They are quite informative. A first conclusion is that the N57A mutation does not abolish all FG interactions. It still allows for some hexamer CA binding, e.g., to Nup58 FG repeats (Fig. 2c) and for a reduced but still prominent capsid targeting to HeLa NPCs (Fig. 4a) as well as to mouse oocyte NPCs when injected into the cytoplasm (fig. 4c). However, the mutant capsid did not reach the nuclear interior. This failed NPC passage, in turn, coincided with a ~15-fold lower partitioning into the GLFG phase, indicating again that the two readouts are intimately linked.

8) For completeness, Fig 3d should be expanded to include capsomeres shown in panel b.

Quantification added as suggested. The partially assembled capsomer reached only a partition coefficient of ~2.5 inside the FG phase, which is 200 times less than the fully assembled capsid spheres. The effect of assembly is thus greater than the images suggest.

9) Page 4 “pointing to the capsid interior”. Is there data to show this? (or publication to cite?).

This is simply concluded from the various published CA capsomer/ capsid structures in which the C terminus of CA always points to the inside of the capsid, so C-terminally fusing a protein (here GFP) to it should result in an interior-facing position. This view is supported by the observation that exchanging GFP for the extremely FG-phobic sinGFP4a variant does not impede FG partitioning of the fully assembled capsid spheres – most likely because sinGFP4a is hidden inside the capsid and shielded from contact with the FG phase.

Minor:

10) ED Fig 4 is cited in text prior to ED Fig 3 (page 6). Please renumber the Figs to reflect the order in which they are cited in main text.

Figure and panel orders have changed again to accommodate new data, however, we have carefully double-checked that we now cite all panels in the appropriate order.

11) Fig 4 Title “Giant” is overly dramatic and should be deleted. These are normal HIV size.

We eliminated the word ‘Giant’ from the figure title as well as another incidence in the discussion.

Cited references

- Frey S, Rees R, Schünemann J, Ng SC, Fünfgeld K, Huyton T, Görlich D (2018) Surface properties determining passage rates of proteins through nuclear pores. *Cell*, **174**: 202–217.e9. doi: 10.1016/j.cell.2018.05.045
- Hülsmann BB, Labokha AA, Görlich D (2012) The permeability of reconstituted nuclear pores provides direct evidence for the selective phase model. *Cell*, **150**: 738–751. doi: 10.1016/j.cell.2012.07.019
- Labokha AA, Gradmann S, Frey S, Hülsmann BB, Urlaub H, Baldus M, Görlich D (2013) Systematic analysis of barrier-forming FG hydrogels from *Xenopus* nuclear pore complexes. *EMBO J*, **32**: 204–218. doi: 10.1038/emboj.2012.302
- Michelitsch MD, Weissman JS (2000) A census of glutamine/asparagine-rich regions: implications for their conserved function and the prediction of novel prions. *Proc Natl Acad Sci U S A*, **97**: 11910–11915. doi: 10.1073/pnas.97.22.11910
- Najbauer EE, Ng SC, Griesinger C, Görlich D, Andreas LB (2022) Atomic resolution dynamics of cohesive interactions in phase-separated Nup98 FG domains. *Nat Commun*, **13**: 1494. doi: 10.1038/s41467-022-28821-8
- Ng SC, Güttler T, Görlich D (2021) Recapitulation of selective nuclear import and export with a perfectly repeated 12mer GLFG peptide. *Nat Commun*, **12**: 4047. doi: 10.1038/s41467-021-24292-5
- Pornillos O, Ganser-Pornillos BK, Banumathi S, Hua Y, Yeager M (2010) Disulfide bond stabilization of the hexameric capsomer of human immunodeficiency virus. *J Mol Biol*, **401**: 985–995. doi: 10.1016/j.jmb.2010.06.042
- Schirra RT, Dos Santos NFB, Zadrozny KK, Kucharska I, Ganser-Pornillos BK, Pornillos O (2023) A molecular switch modulates assembly and host factor binding of the HIV-1 capsid. *Nat Struct Mol Biol*, **30**: 383–390. doi: 10.1038/s41594-022-00913-5
- Schmidt HB, Görlich D (2015) Nup98 FG domains from diverse species spontaneously phase-separate into particles with nuclear pore-like permselectivity. *Elife*, **4**: e04251. doi: 10.7554/eLife.04251
- Zhang Z, He M, Bai S, Zhang F, Jiang J, Zheng Q, Gao S, Yan X, Li S, Gu Y, Xia N (2018) T = 4 icosahedral HIV-1 capsid as an immunogenic vector for HIV-1 V3 loop epitope display. *Viruses*, **10**: 667. doi: 10.3390/v10120667